



# The effects of isoprene and $NO_x$ on secondary organic aerosols

# formed through reversible and irreversible uptake to aerosol water

Marwa M. H. El-Sayed[1], Diana L. Ortiz-Montalvo[2], Christopher J. Hennigan[1]

[1] Department of Chemical, Biochemical and Environmental Engineering, University of Maryland, Baltimore County, Baltimore, MD, USA.
[2] National Institute of Standards and Technology (NIST), Gaithersburg, MD, USA.

*Correspondence to*: Christopher J. Hennigan (hennigan@umbc.edu)

**Abstract.** Isoprene oxidation produces water-soluble organic gases capable of partitioning to aerosol liquid water. The formation of secondary organic aerosols through such aqueous pathways (aqSOA) can take place either reversibly or irreversibly; however, the split between these fractions in the atmosphere is highly uncertain. The aim of this study was to characterize the reversibility of aqSOA formed from isoprene at a location in the eastern United States under substantial influence from both anthropogenic and biogenic emissions. The reversible and irreversible uptake of water-soluble organic gases to aerosol water was characterized in Baltimore, MD using measurements of particulate water-soluble organic carbon ($WSOC_p$) in alternating dry and ambient configurations. $WSOC_p$ evaporation with drying was observed systematically throughout the late spring and summer, indicating reversible aqSOA formation during these times. We show through time lag analyses that $WSOC_p$ concentrations, including the $WSOC_p$ that evaporates with drying, peak ~(6 to 11) h after isoprene concentrations, with maxima at a time lag of 9 h. The absolute reversible aqSOA concentrations, as well as the relative amount of reversible aqSOA, increased with decreasing $NO_x$/isoprene ratios, suggesting that isoprene epoxydiol (IEPOX) or other low-$NO_x$ oxidation products were responsible for these effects. The observed relationships with $NO_x$ and isoprene suggest that this process occurs widely in the atmosphere, and is likely more important in other locations characterized by higher isoprene and/or lower $NO_x$ levels. It is also likely that this phenomenon will increase in importance in the future, given predictions of biogenic and anthropogenic emissions under future regulatory and climate scenarios.

## 1 Introduction

Isoprene (2-methyl-1,3-butadiene, $C_5H_8$) is the most abundant non-methane organic compound emitted globally (Guenther et al., 2012). The oxidation of isoprene has important implications for air quality and climate, as it contributes substantially to ozone and secondary organic aerosol (SOA) formation. In regions with high isoprene emissions, such as the southeastern United States, isoprene is the dominant SOA precursor during summer (Ying et al., 2015; Kim et al., 2015). The oxidation products of isoprene include compounds that partition to aerosol liquid water (ALW), such as isoprene epoxydiol (IEPOX), glyoxal and methylglyoxal. These species do not partition to dry particles (Kroll et al., 2005; Nguyen et al., 2014), so their condensed phase products are called aqueous SOA (aqSOA) (Ervens et al., 2011). IEPOX uptake also depends on the inorganic composition and acidity of the seed



particles (Surratt et al., 2010; Gaston et al., 2014; Budisulistiorini et al., 2017; Lin et al., 2012). A body of work indicates that the uptake of water-soluble organic gases into atmospheric waters (clouds, fogs, and aerosol water) is an important pathway for SOA formation (Ervens et al., 2011). Aerosol water plays a significant role in isoprene SOA formation (Wong et al., 2015), and the majority of isoprene SOA is currently thought to form through aqueous

pathways (Marais et al., 2016). Isoprene emissions show strong seasonal variations in most locations (Guenther et al., 2012), suggesting that aqSOA formation is similarly seasonal in nature. Indeed, SOA formed from IEPOX shows a pronounced seasonal signature in the southeastern U.S. that is consistent with isoprene emissions (Budisulistiorini et al., 2016; Xu et al., 2015).

Although substantial evidence from laboratory, modeling, and ambient studies indicates the importance of aqSOA

formation, many uncertainties remain in understanding this pathway on a mechanistic level (McNeill, 2015). A significant uncertainty is the fate of aqSOA under conditions of water evaporation, such as in a cloud cycle or with diurnal changes in ambient relative humidity (RH). The formation of aqSOA is initiated by the equilibrium (and thus, reversible) partitioning of water-soluble organic gases to liquid water (McNeill, 2015). In the aqueous phase, the dissolved organics can undergo reversible reactions such as hydration and oligomerization (De Haan et al., 2009)

or irreversible reactions such as acid catalysis, reaction with inorganics or radical reactions (e.g. (Ervens et al., 2014; Ortiz-Montalvo et al., 2014; Lee et al., 2013). The former process implies that at least some of the dissolved organics will repartition back to the gas phase when water evaporates, while the latter process can form low-volatility products that remain in the particle phase even after the evaporation of water. Most clouds are non-precipitating (Pruppacher, 1986) and ALW changes throughout the day with changing RH (Nguyen et al., 2014;

Khlystov et al., 2005); thus, determining whether the uptake is reversible or irreversible is critical in understanding the fate of many oxidized organics in the atmosphere. Models predict vastly different amounts of aqSOA depending on whether the uptake of water-soluble organic gases is assumed to be completely irreversible or whether a reversible pathway is also considered (Marais et al., 2016; Pye et al., 2013).

It is important to note that we define aqSOA as all organics present in the condensed phase through partitioning to

liquid water, regardless of whether the uptake is reversible or irreversible. Although some definitions of aqSOA only include the organic material that is taken up into liquid water and remains in the particle phase after water evaporation (e.g. Ervens et al., 2011), we favor a more comprehensive definition since the organics contribute to aerosol effects on health and optical properties when they are in the condensed phase. Our definition is consistent with the treatment of other semi-volatile aerosol species such as ammonium nitrate. It is, however, important to

distinguish reversible and irreversible aqSOA since the atmospheric lifetime of these compounds may differ significantly depending on their phase (Nguyen et al., 2015). Therefore, we define the low-volatility products that remain in the particle phase after the evaporation of liquid water as "irreversible aqSOA", and the organic compounds taken up in liquid water that repartition back to the gas phase with water evaporation as "reversible aqSOA". While ambient studies provide evidence for both reversible and irreversible aqSOA formation (El-Sayed et

al., 2016; El-Sayed et al., 2015), the reasons underlying these differences are still unclear.

Nitrogen oxides ($NO_x \equiv NO + NO_2$) may be one factor affecting the reversibility of isoprene aqSOA. $NO_x$ plays a critical role in the reaction of volatile organic compounds (VOCs). This includes a major effect on isoprene





oxidation chemistry, and on the yield of SOA from isoprene (Kroll and Seinfeld, 2008; Ervens et al., 2008). $NO_x$ affects the volatility, oxidation state, and aging of isoprene-derived SOA (Xu et al., 2014). Recent modeling studies predict that isoprene oxidation in the eastern U.S. is split almost equally between high- and low-$NO_x$ pathways (Travis et al., 2016). Laboratory studies show significant evaporation of aqueous isoprene SOA particles when

dried, indicating reversible aqSOA (Wong et al., 2015). This is consistent with our understanding of aqSOA formed from individual isoprene oxidation products, thought to be predominantly IEPOX and glyoxal (Sareen et al., 2017). Glyoxal is formed from isoprene through low- and high-$NO_x$ pathways (Chan Miller et al., 2017), and is taken up to ALW reversibly and irreversibly (Ortiz-Montalvo et al., 2012; Galloway et al., 2009). IEPOX is formed predominantly through the low-$NO_x$ pathway (Paulot et al., 2009), and its uptake to ALW could be reversible or

irreversible (Nguyen et al., 2014; Riedel et al., 2015). Therefore, potential differences in reversible aqSOA associated with $NO_x$ may be due to differences in IEPOX production under these chemical regimes.

The aim of this study was to characterize the effect of isoprene on the reversibility of aqSOA at a site in the eastern U.S. heavily impacted by biogenic and anthropogenic emissions. The formation of aqSOA from IEPOX is a summertime occurrence in this region (Xu et al., 2015; Budisulistiorini et al., 2016), driven by the seasonality in

isoprene emissions and ALW content (Xu et al., 2017b). However, the lack of specific SOA marker compounds has prevented a more comprehensive approach to measuring seasonal aqSOA from other precursors, such as glyoxal and methylglyoxal. Further, IEPOX taken up reversibly to ALW may not be measured (or may be measured incompletely) by some instruments capable of identifying IEPOX-SOA (El-Sayed et al., 2016). In this study, we determine the impacts of isoprene and $NO_x$ on the reversible and irreversible uptake of water-soluble organic gases

to aerosol liquid water in Baltimore, MD.

## 2 Methods

### 2.1 WSOC measurements

Ambient measurements were carried out across all four seasons in Baltimore, MD (Table 1). The experimental setup has been described in detail elsewhere (El-Sayed et al., 2016; El-Sayed et al., 2015). Briefly, water-soluble

organic carbon was measured in the gas phase ($WSOC_g$) using a mist chamber (MC) and in the particle phase ($WSOC_p$) using a Particle-into-liquid sampler (PILS, Brechtel Manufacturing), both coupled to a total organic carbon (TOC) analyzer operated in Turbo mode (Model 900 Turbo, GE Analytical). The $WSOC_p$ measurement was alternated between an ambient channel ($WSOC_p$) and a 'dried' channel ($WSOC_{p,dry}$) using an automated 3-way valve (Brechtel Manufacturing)[1]. The $WSOC_p$ sample was at ambient RH while the $WSOC_{p,dry}$ sample passed through a

silica gel diffusion dryer (Table S1). Both the $WSOC_p$ and the $WSOC_{p,dry}$ samples pass through a parallel-plate carbon denuder prior to sampling in the PILS. This reduces gas-phase interferences, which are minor in the PILS (Sullivan et al., 2004), and prevents the re-condensation of volatilized organic gases that evaporate in the dryer. Although some gas-phase organics may be lost to the silica gel (Faust et al., 2017), potentially perturbing the gas-

---

[1] Certain commercial equipment, instruments or materials are identified in this paper to foster understanding. Such identification does not imply recommendation or endorsement by the National Institute of Standards and Technology, nor does it imply that the materials or equipment identified are necessarily the best available for the purpose.



particle equilibrium for the dry channel, sampling both channels through the carbon denuder should minimize such differences. Further, based upon the timescales of ambient organic aerosol (OA) equilibration (minutes-to-hours) (Saha et al., 2017), it is highly unlikely that stripping gas-phase compounds would produce any appreciable OA evaporation with only the 7 s residence time encountered in our system. The diffusion dryer does not implement

heating, so differences in the $WSOC_p$ concentrations between the two channels are due to $WSOC_p$ evaporation that results from ALW evaporation. $WSOC_p$ losses through the 3-way valve and through the dried channel are less than 1 % (mass concentration basis) (El-Sayed et al., 2016): no corrections to the data were applied. A ratio of OM/OC=2.1 was used to convert aerosol organic carbon (OC) into organic mass (OM), based upon characterizations of $WSOC_p$ in the eastern U.S. (Xu et al., 2017a).

The fully-automated online system was housed in a temperature-controlled environmental enclosure (EKTO, Inc.) placed on the rooftop of the Engineering Building at the University of Maryland, Baltimore County (UMBC). The three samples: $WSOC_g$, $WSOC_p$ and $WSOC_{p,dry}$ were repeatedly measured in a 14-min cycle with sampling times of 4 min, 5 min and 5 min, respectively. Dynamic blanks were measured regularly throughout each ambient sampling period.

**2.2 VOC and $NO_x$ measurements**

Isoprene measurements from the Essex Photochemical Assessment Monitoring Stations (PAMS) (AQS ID# 240053001) were provided by Maryland Department of the Environment (MDE). The Essex site represents the PAMS station closest to UMBC (~20 km distance). Isoprene was measured by MDE every six days from September to May, and hourly during the summer (June, July and August). The hourly isoprene measurements were automated

following EPA method 142, using cryogenic preconcentration for sample collection followed by analysis via gas chromatography with flame ionization detection (GC-FID, Perkin Elmer Clarus 500). Hourly measurements of $NO_x$ ($NO_x$=NO+$NO_2$) concentrations were also carried out by MDE at the Essex site following method 74 (chemiluminescence). Data were acquired from the U.S. Environmental Protection Agency (https://aqs.epa.gov/api).

**3 Results**

An overview of the seasonal sampling periods is given in Table 1. Measurements were taken from 3 to 4 weeks on average during each of the four seasons. Note that the spring season has been divided into early (23 April to 8 May) and late (9 May to 14 May) periods due to the differences in the $WSOC_p$ results observed during these two periods.

The $WSOC_p$ measurements has been reported to be a good surrogate of the total SOA in the atmosphere

(Weber et al., 2007; Kondo et al., 2007), which includes aqSOA as well as SOA formed

through traditional gas-phase partitioning (Donahue et al., 2009). In this regard, we first identified periods when aqSOA was formed. The formation of aqSOA has been observed throughout the year, except for the early spring season. This observation was based on the relationship between the fraction of total WSOC in the particle phase, $F_p$ ($F_p$ = $WSOC_p$/($WSOC_p$ + $WSOC_g$)) as a function of RH in combination with

seasonal ALW analyses. The individual results for the fall and summer have been previously reported (El-Sayed et




al., 2016; El-Sayed et al., 2015). The seasonal results are the subject of ongoing analyses that presents a synthesis of aqSOA formation across all seasons.

### 3.1 Reversibility of aqSOA formation by season

Previous studies conducted by our group have provided evidence for both irreversible (El-Sayed et al., 2015) and

reversible (El-Sayed et al., 2016) aqSOA formation during the fall and summer seasons, respectively. Figure 1 shows the $WSOC_{p,dry}/WSOC_p$ ratio across all of the seasons. A ratio of unity indicates that drying did not impact $WSOC_p$ while a ratio less than unity indicates that particle drying caused the evaporation of some $WSOC_p$, and thus was considered reversible aqSOA (El-Sayed et al., 2016). Fig. 1 shows that the $WSOC_{p,dry}/WSOC_p$ ratio was unity during the fall and winter, indicating that the $WSOC_p$ remained in the condensed phase upon drying. Therefore, the

aqSOA formation that was observed occurred irreversibly (El-Sayed et al., 2015). In the early spring, the $WSOC_{p,dry}/WSOC_p$ ratio was also unity, but this was expected since no $F_p$-RH enhancement was observed and no significant aqSOA was observed during this period. Beginning in the late spring and continuing into the summer, the $WSOC_{p,dry}/WSOC_p$ ratio was systematically lower than unity. During both seasons, we observed systematic evaporation of some $WSOC_p$ as a result of the ALW evaporation. In the late

spring, the $WSOC_{p,dry}/WSOC_p$ ratio was 0.92, on average, and decreased further during the summer where it reached an average of 0.87 (El-Sayed et al., 2016). This observation indicates that at least some of the aqSOA formation occurring in the late spring and summer seasons was reversible. $WSOC_p$ evaporation was higher during the night than during the day (Fig. S1), likely due to higher RH levels and higher ALW at night (Guo et al., 2015). The purpose of this study is to characterize the reasons underlying the seasonal differences in $WSOC_{p,dry}/WSOC_p$ shown

in Fig. 1.

### 3.2 Climatology of isoprene

Isoprene oxidation products are thought to be the most important precursors to aqSOA formation (Marais et al., 2016). Figure 2 shows the average annual climatology of isoprene in Baltimore, MD. These measurements were made at the MDE Essex site, a location ~20 km from UMBC where the WSOC measurements were conducted. In

the eastern U.S., isoprene emissions are regional (Palmer et al., 2003); therefore, data from the Essex site will show consistent trends with those at UMBC. Isoprene concentrations in Baltimore tend to be very low in the winter and early spring seasons, with average monthly values of ~0.2 ppbC, but they start to rise sharply at the beginning of May, and remain elevated (though variable) during the summer season. This is highly consistent with previously measured seasonal isoprene emissions in other parts of the eastern U.S. (Goldstein et al., 1998). Isoprene

concentrations decrease dramatically in September (average decrease of 70 % from 1 Sept to 30 Sept), and then remain low through the winter.

### 3.3 Effect of isoprene on reversible aqSOA

During the late spring, our observation of the onset of reversible aqSOA formation corresponds to the dramatic increase in isoprene concentrations (Fig. 2). Observations of the AMS IEPOX factor in the southeastern U.S. show

similarly sharp transitions in the spring and fall (Budisulistiorini et al., 2016). The highest reversible aqSOA levels were observed during the summer when isoprene emissions were at their maximum.




Due to the magnitude of regional isoprene emissions and its predicted contribution to SOA, we would expect relationships between isoprene and both $WSOC_g$ and $WSOC_p$ concentrations. However, simple correlations between isoprene and WSOC are not expected, due to dramatic differences in their atmospheric lifetimes. Under typical summertime conditions, the oxidation of isoprene to form $WSOC_g$ will take a few hours (Hodzic et al., 2014). These

oxidation products can undergo further reactions to form lower volatility compounds that partition to the aerosol phase contributing to $WSOC_p$, a process that is expected to take several hours (Ng et al., 2006; Atkinson and Arey, 2003).

The relationship between isoprene and WSOC (both $WSOC_p$ and $WSOC_g$) was characterized for the summer, when hourly isoprene data were available. To account for the differences in the expected timeframe for

transformation of isoprene into $WSOC_g$ and $WSOC_p$, we analyzed the WSOC concentrations as a function of isoprene with a variable time lag. We investigated the relationship between the isoprene concentrations at time t and the WSOC concentrations at t + n, where n is the time lag, which was systematically varied from (0 to 13) h. For example, a 1 h time lag indicates that the isoprene concentrations at t are compared to the WSOC concentrations measured t + 1 h after those of isoprene. An offset of zero indicates that the timing of the WSOC measurements is

aligned with the timing of the isoprene measurements. During each hour, there were 4 to 5 $WSOC_p$ and $WSOC_g$ measurements corresponding to one isoprene sample; therefore, hourly averages of WSOC were calculated to provide a consistent basis for analysis.

First, the isoprene-$WSOC_g$ relationship was analyzed for time lags in the range of (0 to 6) h (Fig. 3). The $WSOC_g$ data were binned based on the corresponding isoprene concentrations; each marker represents the median of the

$WSOC_g$ concentration within each isoprene concentration bin. At (0 to 2) h time lags, no relationship was observed between isoprene concentrations and $WSOC_g$. This was anticipated because isoprene has a typical atmospheric lifetime of (1 to 2) hours against oxidation by OH (Atkinson and Arey, 2003). However, with a time lag of 3 h, an increase in isoprene concentrations was coincident with a significant increase in $WSOC_g$ concentrations. This effect was observed for time lags up to 5 h, as illustrated by the solid blue lines in Fig. 3. Across the entire summer, a 5

ppbC increase in isoprene concentrations was associated with a median increase of 2.0 $\mu$g-C m$^{-3}$ in $WSOC_g$. When the time lag between isoprene and $WSOC_g$ was more than 5 h, there was no longer a relationship between isoprene and $WSOC_g$ concentrations. This observation highlights the effect of isoprene on the formation of water-soluble organic gases. Isoprene and $WSOC_g$ showed similar diurnal profiles during the summer, especially when the time lag was considered (Fig. S2). Overall, this suggests that fresh isoprene emissions take about (3 to 5) h to form

$WSOC_g$ in an urban environment during typical summertime conditions. Note that the measurement of $WSOC_g$ only includes compounds with effective Henry's law constants above $\sim10^3$ M/atm (Spaulding et al., 2002), so the MC does not efficiently sample many first-generation isoprene oxidation products, such as methacrolein ($K_H = 4 \times 10^0$ M/atm) or methyl vinyl ketone ($K_H = 4 \times 10^1$ M/atm) (Sander, 2015).

The relationship between isoprene and evaporated $WSOC_p$ (i.e., reversible aqSOA) was characterized using the

same time-lag analysis, extended from n = (0 to 13) h. At time lags less than 5 h, there was no relationship between isoprene and evaporated $WSOC_p$ concentrations (red dotted lines in Fig. 4). However, the amount of evaporated $WSOC_p$ increased significantly with increasing isoprene concentrations when the evaporated $WSOC_p$ time lag was





in the range of (6 to 11) h (green solid lines in Fig. 4). The highest response of evaporated $WSOC_p$ to isoprene was found for a time lag of 9 h (Fig. S3). At this time lag, an increase of 5 ppbC in isoprene concentrations led to a median increase of ~0.7 µg m$^{-3}$ in evaporated $WSOC_p$. Beyond the 11 h time lag, no relationship was observed between isoprene and evaporated $WSOC_p$ levels (blue dotted lines in Fig. 4). The average evaporated $WSOC_p$

concentrations showed a similar increase with increasing isoprene concentrations, but were even higher than the median levels (Fig. S3). For example, at a 9 h time lag, a 5 ppbC increase in isoprene corresponded to an average increase in evaporated $WSOC_p$ of ~1.6 µg m$^{-3}$. The (6 to 11) h time lag between isoprene and the evaporated $WSOC_p$ is highly consistent with recent model predictions of IEPOX SOA formation in the eastern U.S. (Budisulistiorini et al., 2017). This observed (6 to 11) h time lag between isoprene and the evaporated $WSOC_p$ is

likely due to multi-generational oxidation (Carlton et al., 2009; Hodzic et al., 2014; Paulot et al., 2009). Alternately, it could be that the isoprene oxidation products that partition reversibly to liquid water were formed relatively quickly (< 6 h), but responded to the diurnal cycle in ALW, which peaks in the eastern U.S. in the early morning hours (Guo et al., 2015). The observed delay time could also be the combination of these factors. Consistent with Fig. 3 and Fig. 4, there was also a strong relationship between the $WSOC_g$ concentration and the time-offset

evaporated $WSOC_p$ concentration (Fig. S4). The above observations suggest that isoprene is strongly linked with the formation of reversible aqSOA in the eastern U.S. Based on this relationship, we next consider the effect of $NO_x$ on reversible aqSOA formation since $NO_x$ is critical to isoprene oxidation chemistry (Kroll et al., 2006).

### 3.4 Effect of $NO_x$ on reversible aqSOA

Figure 5 shows the relationship between evaporated $WSOC_p$ and the $NO_x$/isoprene ratio during the summer. For

this analysis, hourly $NO_x$/isoprene ratios and the hourly evaporated $WSOC_p$ concentrations with a 9 h time lag were used. Blue markers represent the mean of the evaporated $WSOC_p$ concentrations within each $NO_x$/isoprene bin. It is clear from Fig. 5 that the amount of evaporated $WSOC_p$ decreased dramatically with an increase in $NO_x$/isoprene ratios. At low $NO_x$/isoprene ratios (less than 0.5 ppb/ppbC), the amount of evaporated $WSOC_p$ was at its maximum (average of 1.4 µg m$^{-3}$), however at $NO_x$/isoprene ratios more than 15 ppb/ppbC, the evaporated $WSOC_p$ was as low

as 0.2 µg m$^{-3}$. Generally, the evaporated $WSOC_p$ decreased significantly with the increase in $NO_x$/isoprene ratios, but flattened out beyond $NO_x$/isoprene ratios of ~5 ppb/ppbC.

Similarly, the effect of $NO_x$/isoprene ratios on $WSOC_p$ concentrations during the summer is shown in Fig. 6. As in Fig. 5, the hourly $NO_x$/isoprene ratios were compared against the hourly $WSOC_p$ concentrations at a time lag of 9 h. At $NO_x$/isoprene ratios of less than 0.5 ppb/ppbC, the average $WSOC_p$ concentration was ~5 µg m$^{-3}$, but it

decreased substantially to ~1.5 µg m$^{-3}$ (almost summertime $WSOC_p$ background levels) at $NO_x$/isoprene ratios above 15 ppb/ppbC.

Although isoprene emissions decrease dramatically during September, there are still periods with elevated concentrations. If isoprene is indeed associated with the evaporated $WSOC_p$ that we observed during the late spring and summer, then a logical question is why we did not observe this phenomenon during measurements throughout

September (Fig. 1, El-Sayed et al., 2015). Here, we analyze the effects of $NO_x$ and isoprene on the reversibility of isoprene aqSOA by considering the average daily $NO_x$/isoprene ratios during the late spring, summer, and fall. For this analysis, daily averages were used due to the lack of hourly isoprene measurements during the late spring and



fall. The relationship between the $WSOC_{p,dry}/WSOC_p$ and $NO_x$/isoprene ratios across all three seasons is shown in Fig. 7. Figures 5 and 6 show that the relationships of the $NO_x$/isoprene ratio with $WSOC_p$ and evaporated $WSOC_p$ are qualitatively similar. However, it is clear from Fig. 7 that $WSOC_p$ and the evaporated $WSOC_p$ are affected differently by $NO_x$/isoprene. The days in which average $NO_x$/isoprene ratios were higher than ~5 ppb/ppbC were

characterized by $WSOC_{p,dry}/WSOC_p$ ratios very close to unity, indicating irreversible aqSOA. On the other hand, the days in which $NO_x$/isoprene ratios were lower than ~5 ppb/ppbC were all characterized by $WSOC_{p,dry}/WSOC_p$ ratios lower than unity, indicating some reversible aqSOA on these days. Further, the $WSOC_{p,dry}/WSOC_p$ ratio decreased with decreasing $NO_x$/isoprene ratios under this condition. IEPOX is produced under low-$NO_x$ conditions with very limited formation in $NO_x$ rich environments (Zhang et al., 2017) whereas glyoxal can be produced from both low-

and high-$NO_x$ pathways with higher yields at high-$NO_x$ conditions (Chan Miller et al., 2017). Based on our observations, this suggests that IEPOX was more abundant during the late spring and summer and was responsible for the reversible aqSOA formed under the lower $NO_x$/isoprene conditions. These results provide an explanation for the variability in the seasonal occurrence of reversible aqSOA in the eastern U.S.

## 4 Atmospheric Implications

These results represent, to our knowledge, the first observations to characterize the seasonal occurrence of reversible aqSOA formation. The results have important implications for aerosol measurements that implement drying, which may not measure (or may incompletely measure) reversible aqSOA. Our results suggest that this is especially important in areas with high isoprene emissions. For example, (Zhang et al., 2012) observed substantial loss of $WSOC_p$ (~30 % on average) from Federal Reference Method (FRM) filters in the southeastern U.S. It is

likely that reversible aqSOA contributed to this measurement artifact, although direct comparisons to our $WSOC_{p,dry}$ measurement would be needed to confirm this hypothesis. These compounds are important, since they contribute to aerosol effects – visibility, AOD, health, climate – when they are in the condensed phase.

The effect of water evaporation on $WSOC_p$ also has important implications for the representation of SOA formation in models. Models that include aqSOA and aerosol multiphase chemistry can improve predictions of OA

(e.g., Carlton et al., 2008; Marais et al., 2016). However, accounting for both reversible and irreversible uptake of water-soluble organic gases is critical (McNeill, 2015). Comparisons of modeled OA concentrations to ambient measurements may not reveal a problem if the measurements, themselves, are subject to the bias discussed above. Consistent with laboratory studies (Faust et al., 2017), our observations suggest that treatment of aqSOA as an irreversible uptake process is not consistent with actual phenomena occurring in the atmosphere, especially in the

eastern U.S. Although our observations are likely due to a different mechanism, Liu et al. (2016b) and Riva et al., (2017) showed that reactive uptake of isoprene oxidation products can form semi-volatile compounds that re-partition back to the gas phase. Note that the measurement of $WSOC_p$ includes compounds formed through uptake to aqueous particles (aqSOA) and compounds formed through traditional SOA pathways. Therefore, the results in Fig. 1 suggest that although ~10 to 15 % of the total $WSOC_p$ evaporates with drying during the late spring and

summer, the reversible aqSOA fraction is much higher. Further, the fraction of $WSOC_p$ that evaporates with drying is variable, and ranges from 0 to 60 % for individual measurements (El-Sayed et al., 2016). Together, these results





indicate that representations of aqSOA formation through irreversible uptake schemes are not consistent with actual atmospheric phenomena.

The lifetime of organic compounds in the atmosphere is strongly dependent on their phase (Pye et al., 2017). Oxygenated organic compounds in the gas-phase often have much shorter lifetimes than particle-phase organics due to significantly higher dry deposition velocities (Nguyen et al., 2015) and photolysis rates (Fu et al., 2008). Thus, the reversible uptake of $WSOC_g$ to aerosol water may effectively shield these species from such loss processes, resulting in enhanced transport. This may further challenge model predictions of water-soluble organic gases.

We hypothesize that the evaporation of $WSOC_p$ observed with drying during the late spring and summer is due to the reversible partitioning of IEPOX to aerosol water, or to other low-$NO_x$ isoprene oxidation products such as multifunctional hydroperoxides (Liu et al., 2016a; Krechmer et al., 2015; Riva et al., 2016). This is supported by strong associations between the evaporated $WSOC_p$ and isoprene concentrations using the time lag analysis. It is further supported by the decreasing $WSOC_{p,dry}/WSOC_p$ ratios with decreasing $NO_x$/isoprene ratios. Laboratory studies have found reversible and irreversible uptake of IEPOX to aqueous particles (Nguyen et al., 2014; Riedel et al., 2015). However, ambient studies generally suggest that IEPOX-SOA has very low volatility (Lopez-Hilfiker et al., 2016; Hu et al., 2016). This could be due to challenges measuring the reversible aqSOA by the methods used to derive volatilities. For example, it is unclear how the instruments employed by Lopez-Hilfiker et al. (2016) and Hu et al. (2016) respond to unreacted IEPOX present in the aqueous phase. It could also be that the evaporated $WSOC_p$ we observe is contributed by other low-$NO_x$ isoprene oxidation products. Approximately 30 % of isoprene-SOA generated under $NO_x$-free conditions partitioned reversibly to aerosol water, but the molecular identities of the reversible aqSOA were not determined (Wong et al., 2015). Although the experiments of Wong et al., (2015) were performed in a chemical regime where IEPOX formation is favored, it did not contribute to the SOA in their experiments due to high OH levels. The uptake of other, non-IEPOX, low-$NO_x$ oxidation products may explain such observations (Riva et al., 2016; Liu et al., 2016a). Overall, identifying the molecular composition of the reversible aqSOA that is associated with low-$NO_x$ isoprene oxidation will require targeted field measurements.

$NO_x$ plays a critical role in the oxidation of VOCs, including effects on the composition and quantity of SOA produced. Herein, we show that $NO_x$ strongly affects the amount and nature of SOA produced in an urban area that is under substantial influence from biogenic emissions. Higher concentrations of $WSOC_p$ were associated with decreasing $NO_x$/isoprene ratios. The fraction of $WSOC_p$ that evaporated with drying was also inversely related to $NO_x$/isoprene. In the future, isoprene concentrations are predicted to increase in response to changes in temperature and land use associated with climate change (Heald et al., 2008; Sanderson et al., 2003). The eastern U.S. is currently undergoing a transition from high- to low-$NO_x$ chemical regimes (Travis et al., 2016; Edwards et al., 2017), and $NO_x$ levels are likely to continue decreasing (He et al., 2013). This suggests future $NO_x$/isoprene ratios will generally decrease across the eastern U.S., as well, resulting in increased production of reversible aqSOA. The current results are from the greater Baltimore metropolitan area; although we observe a range of $NO_x$ concentrations and $NO_x$/isoprene ratios, these measurements are representative of an urban environment. Thus, we may expect $WSOC_{p,dry}/WSOC_p$ ratios to be even lower in more rural environments impacted by isoprene emissions.

## 5 Uncertainties



Our $WSOC_{p,dry}$ measurement system employs a total drying time of ~7 s. The residence time for equilibrium to take place in evaporating water/organic droplets is dependent on the specific organics as well as the aerosol inorganic chemical composition. Studies that investigated the evaporation of aqSOA formed from reversible IEPOX uptake to aqueous particles found that ~3 s was enough drying time to observe complete evaporation of these

compounds (Nguyen et al., 2014). For glyoxal and glycolaldehyde particles, Ortiz-Montalvo et al., (2014) used a post-drying time of 6 s. Conversely, other studies required up to 20 s for dried succinic acid-ammonium sulfate particles to equilibrate (Yli-Juuti et al., 2013) and up to 2 min for the equilibration of aqueous carbonyl-ammonium sulfate particles that had undergone drying (Galloway et al., 2014). Longer drying times may increase the amount of evaporated aqSOA in our system, indicating that our measurements provide a conservative (low) bound estimate on

the concentration of reversible aqSOA and on the $WSOC_{p,dry}/WSOC_p$ ratio.

Although ALW can affect the gas-particle partitioning of SOA formed through gas-phase oxidation and partitioning (Chang and Pankow, 2010), we believe that the observed $WSOC_p$ evaporation during the late spring and summer seasons was the result of reversible aqSOA. The effect of ALW on gas-particle partitioning is more pronounced at low organic concentrations (1 to 2 $\mu g\ m^{-3}$), and its sensitivity becomes less profound at higher OA

concentrations (Pankow, 2010). Our observations show that the evaporated $WSOC_p$ concentrations increased significantly with an increase in OA concentrations (El-Sayed et al., 2016). Further, the semi-volatile organic compounds most influenced by this water effect are predicted to be the less oxidized, fresh SOA (Pankow, 2010). $WSOC_p$ is much more strongly correlated with LV-OOA compared to SV-OOA (Sun et al., 2011; Xu et al., 2017b; Kondo et al., 2007). Combined with the seasonality of the observed phenomenon, this suggests that the evaporation

of $WSOC_p$ was not due to the overall effects on OA partitioning (Jathar et al., 2016), but was due to the reversible partitioning of water-soluble organic gases to aerosol water.

There is uncertainty in linking the $NO_x$/isoprene ratios to WSOC concentrations. There are differences in the spatial distributions of WSOC, $NO_x$ and isoprene as a result of spatially segregated emissions (Yu et al., 2016) and different atmospheric lifetimes. Wolfe et al., (2016) provided empirical constraints on the effects of $NO_x$ on

formaldehyde (HCHO) yields from isoprene. They dealt with this problem by defining different HCHO background levels for different air masses. In our case, we implemented the variable time lag analysis to show that $WSOC_g$ and $WSOC_p$ responded to $NO_x$/isoprene ratios in a way that is consistent with laboratory and modeling studies. Our analyses are supported by predictions that the chemistry occurring on small scales, such as in individual power plant plumes, does not significantly affect the isoprene-$NO_x$ regime (high- or low-$NO_x$) on regional scales (Yu et al.,

2016). However, factors beyond the $NO_x$/isoprene ratio affect reversible aqSOA formation. The most important factors identified previously include RH and the total $WSOC_p$ concentration (El-Sayed et al., 2016), as well as inorganic aerosol composition (Nguyen et al., 2014). This contributes to the scatter in individual hourly measurements (Fig. S3). Our conclusions are based on average (and median) relationships observed over many weeks and thus capture the prevailing effects of $NO_x$ and isoprene on reversible aqSOA. There is also uncertainty in

the absolute $NO_x$/isoprene ratio that represents the transition to reversible aqSOA. Essex, MD is a suburban site that is located immediately downwind of the Baltimore area. Even though the isoprene-$NO_x$ chemical regime in the eastern U.S. is generally well-represented with model resolution of 28 x 28 km (Yu et al., 2016), a site such as Essex



that is under such heavy urban influence likely has higher $NO_x$ and lower absolute isoprene levels than other sites in the region. Therefore, although Figure 7 suggests that reversible aqSOA formation occurs at $NO_x$/isoprene ratios below 5 ppb/ppbC, transitions at lower ratios may be observed in other areas.

In our study, we assume that the $WSOC_p$ measurement is a surrogate for SOA (Weber et al., 2007). However,
$WSOC_p$ is weakly correlated with lightly-oxygenated components in OA, such as the SV-OOA factor often resolved by the Aerodyne aerosol mass spectrometer (Timonen et al., 2013). Thus, our analysis would likely be a poor method for some SOA systems, for example α-pinene ozonolysis (Jimenez et al., 2009). As discussed above, the $WSOC_g$ measurement does not efficiently sample compounds with low Henry's law constants, including some first generation isoprene oxidation products (Hodzic et al., 2014). These measurement limitations contribute to the (6 to
11) h and (3 to 5) h time lags for the isoprene associations with evaporated $WSOC_p$ and $WSOC_g$, respectively. For many compounds, multi-generation oxidation contributes significantly to SOA formation (Ng et al., 2006), and this is almost certainly the case for atmospheric SOA (Jimenez et al., 2009). However, shorter lag times may be observed with other instruments sensitive to early-generation oxidation products. Finally, our observations of aqSOA are taken near the ground (~20 m), so they do not account for compounds, such as methylglyoxal, that likely
only form aqSOA in clouds (Lim et al., 2013; Sareen et al., 2017).

## 6 Conclusions

Biogenic isoprene emissions affect air quality and climate. The eastern U.S. is undergoing a transition from a high- to –low-$NO_x$ chemical regime, which has broad implications for nighttime chemistry, ozone production, and SOA formation (Travis et al., 2016; Marais et al., 2016; Edwards et al., 2017). Using a time lag analysis, we show
that $NO_x$/isoprene strongly affects concentrations of SOA in the eastern U.S., including SOA formed through the reversible uptake of water-soluble organic gases to aqueous particles. Lower $NO_x$ leads to increased SOA production, and to a higher fraction of aqueous SOA formed reversibly. Our measurements from an urban area suggest that this process is even more important in other, more rural environments. Predictions of future emissions in response to regulations, technology, and climate change also suggest that this process will increase in importance
going forward.

We hypothesize that IEPOX uptake to aqueous particles is responsible for the reversible aqSOA, but other low-$NO_x$ isoprene oxidation products are possible, as well (Wong et al., 2015). We quantify reversible aqSOA through observations of $WSOC_p$ evaporation that results from drying. Ultimately, molecular composition measurements made concurrently with our WSOC system are required to identify the chemical species responsible for this
phenomenon. The evaporation of $WSOC_p$ with drying occurred systematically during the late spring and summer, and was linked to isoprene and $NO_x$. This has importance for a wide range of aerosol measurements that implement drying. It also has importance for modeling multi-phase SOA formation, as simplified treatment of irreversible uptake does not represent actual atmospheric processes.

**Acknowledgments**



The data used in this analysis are available upon request. This work was supported by the National Science
Foundation through Award CHE-1454763.

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



**Table 1.** Seasonal sampling periods in Baltimore, MD.

| Season | Sampling Period |
| --- | :---: |
| **Fall** | 3 – 30 September 2014 |
| **Winter** | 4 February – 23 March 2015 |
| **Early spring** | 23 April – 8 May 2015 |
| **Late spring** | 9 May – 14 May 2015 |
| **Summer** | 6 July – 14 August 2015 |



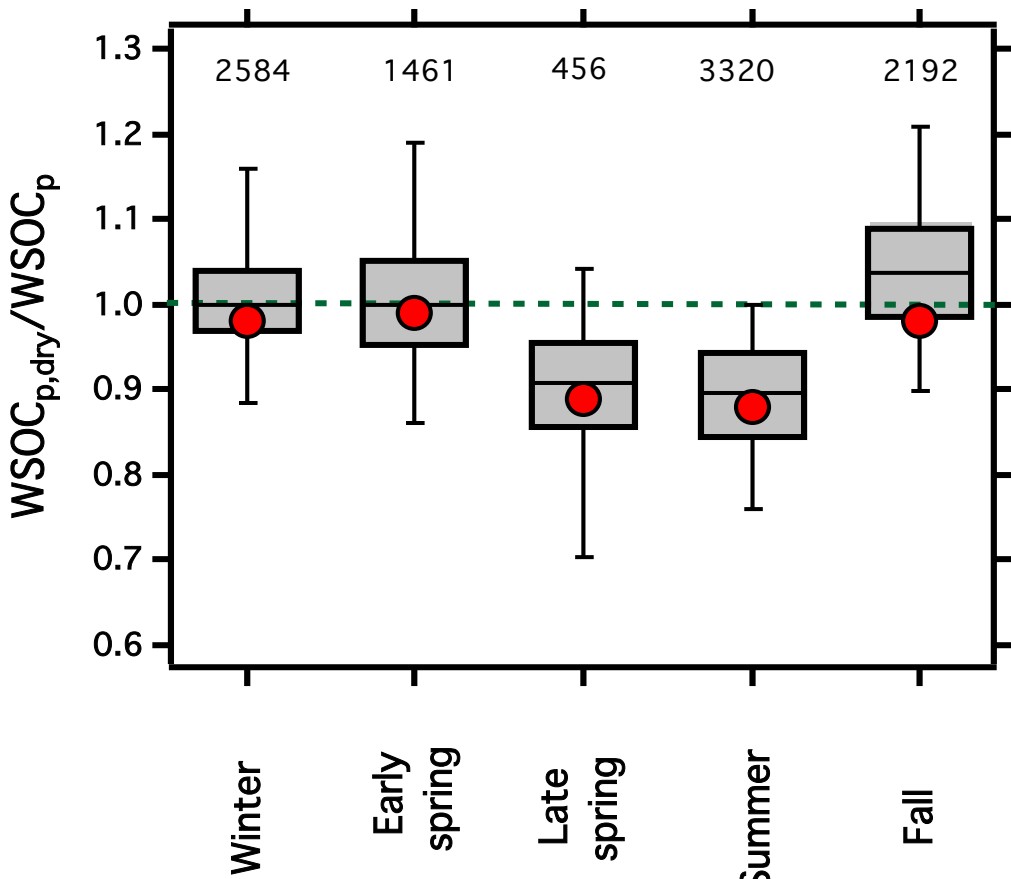

**Figure 1:** Boxplot of the overall seasonal $WSOC_{p,dry}/WSOC_p$ ratios. For each bin, mean (red marker), median (horizontal black line), 25th and 75th percentiles (lower and upper box values), as well as 5th and 95th percentiles (vertical lines) are shown. The dotted green line at unity is shown for visual reference. Numbers at the top represent the number of paired $WSOC_{p,dry}/WSOC_p$ measurements within each season.





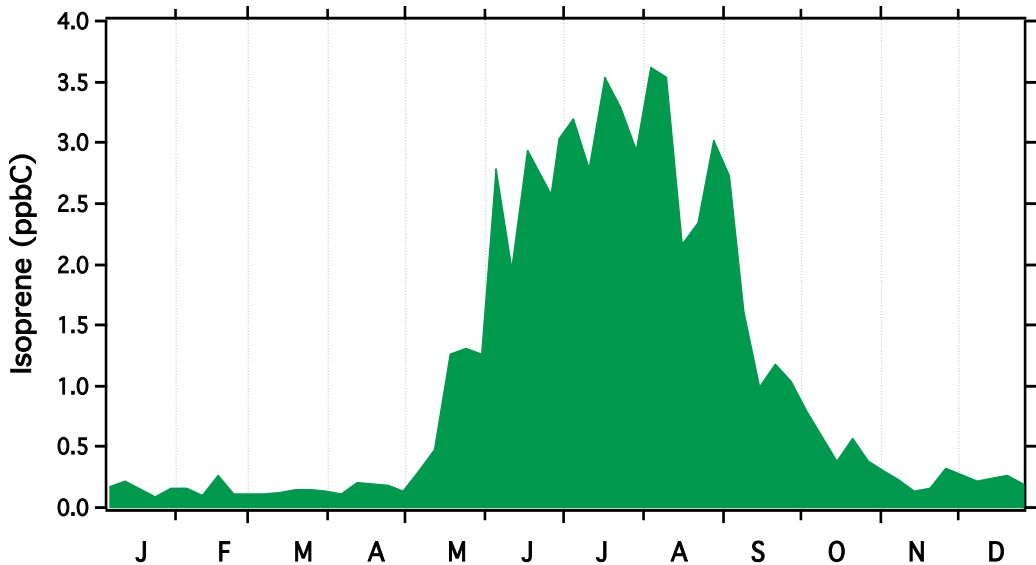

**Figure 2:** Annual average climatology of isoprene concentrations in Essex, MD (2011 to 2015).





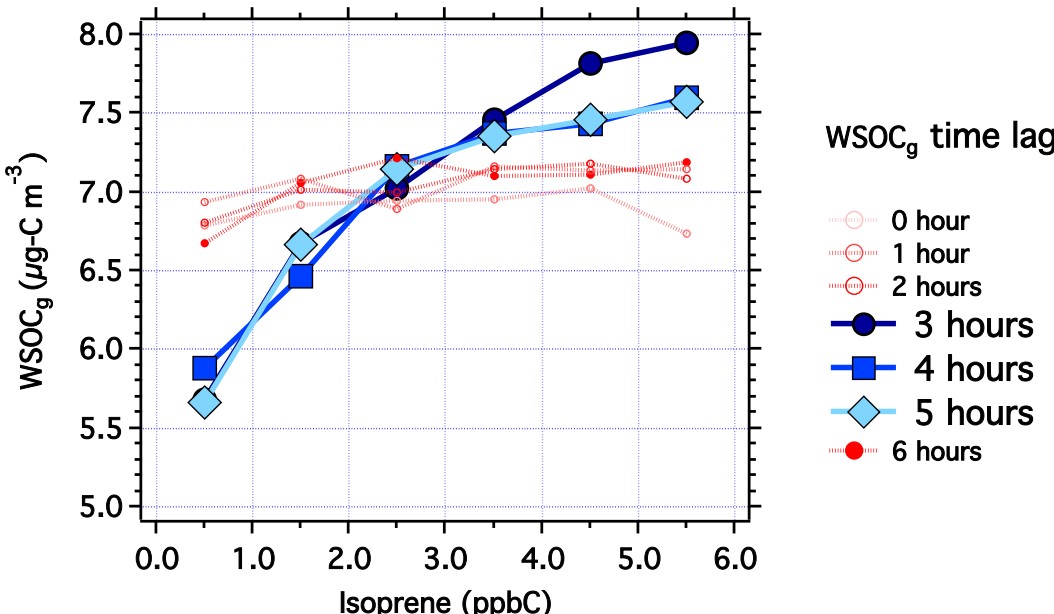

**Figure 3:** Median WSOC$_g$ concentrations as a function of isoprene concentrations at different WSOC$_g$ time lags during the summer. The following isoprene concentrations bins were defined: < 1 ppbC, (1 to 2) ppbC, (2 to 3) ppbC, (3 to 4) ppbC, (4 to 5) ppbC, and > 5 ppbC.



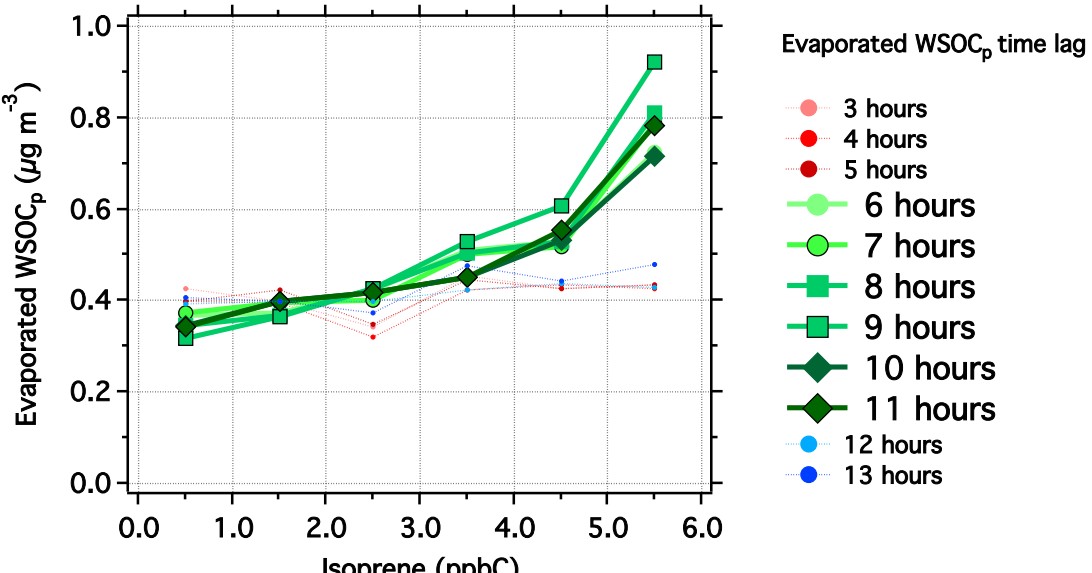

**Figure 4:** Median evaporated WSOC$_p$ concentrations as a function of isoprene concentrations at different evaporated WSOC$_p$ time lags during the summer.



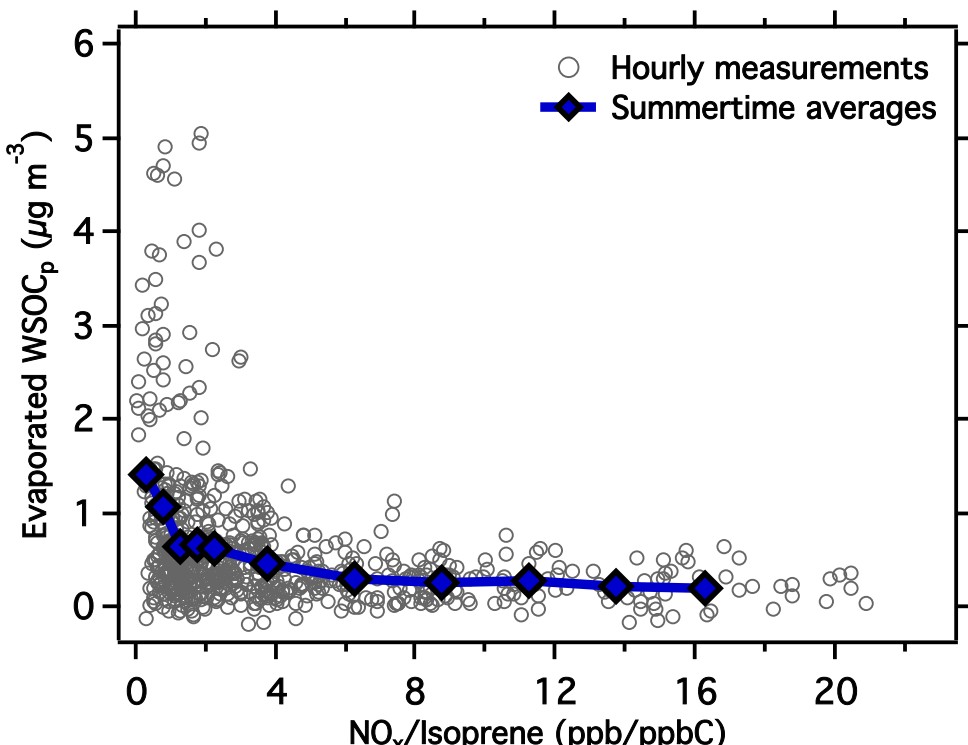

**Figure 5:** Scatter plot of evaporated WSOC$_p$ (9 h time lag) as a function of NO$_x$/isoprene ratio in the summer. Blue diamonds represent the mean at each NO$_x$/isoprene bin. Data were binned according to the NO$_x$/isoprene ratios: bins were defined as < 0.5 ppb/ppbC, (0.5 to 1) ppb/ppbC, (1 to 1.5) ppb/ppbC, (1.5 to 2) ppb/ppbC, (2 to 2.5) ppb/ppbC, (2.5 to 5) ppb/ppbC, (5 to 7.5) ppb/ppbC, (7.5 to 10) ppb/ppbC, (10 to 12.5) ppb/ppbC, (12.5 to 15) ppb/ppbC and > 15 ppb/ppbC.



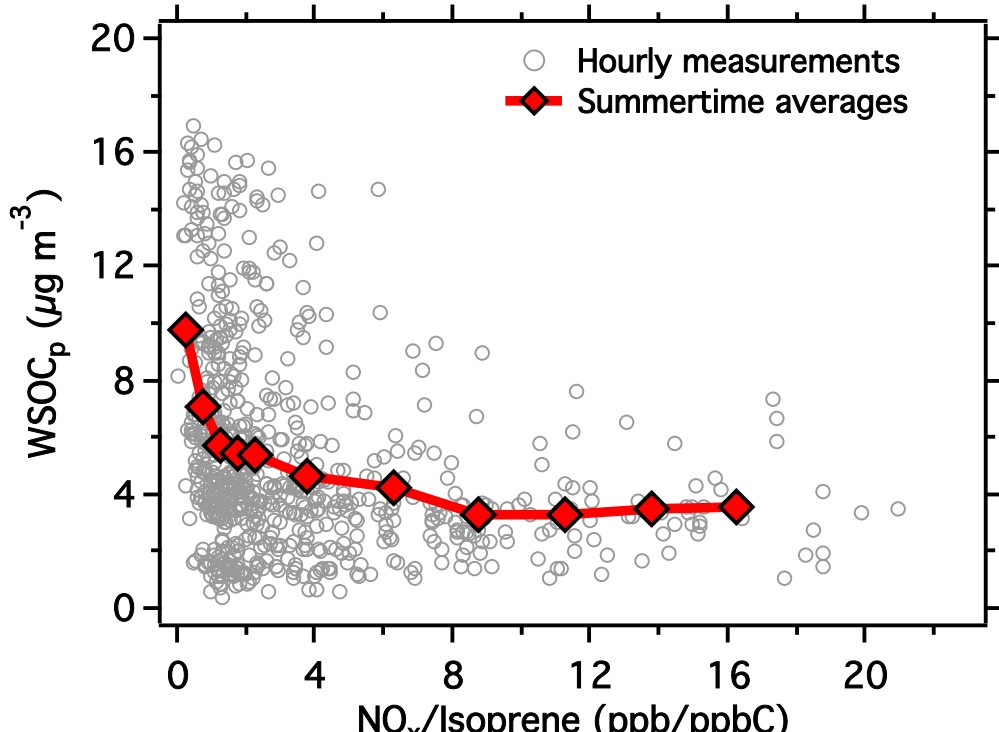

**Figure 6:** Scatter plot of $WSOC_p$ as a function of $NO_x$/isoprene ratio in the summer. Red diamonds represent the mean at each $NO_x$/isoprene bin, defined as in Fig. 5.




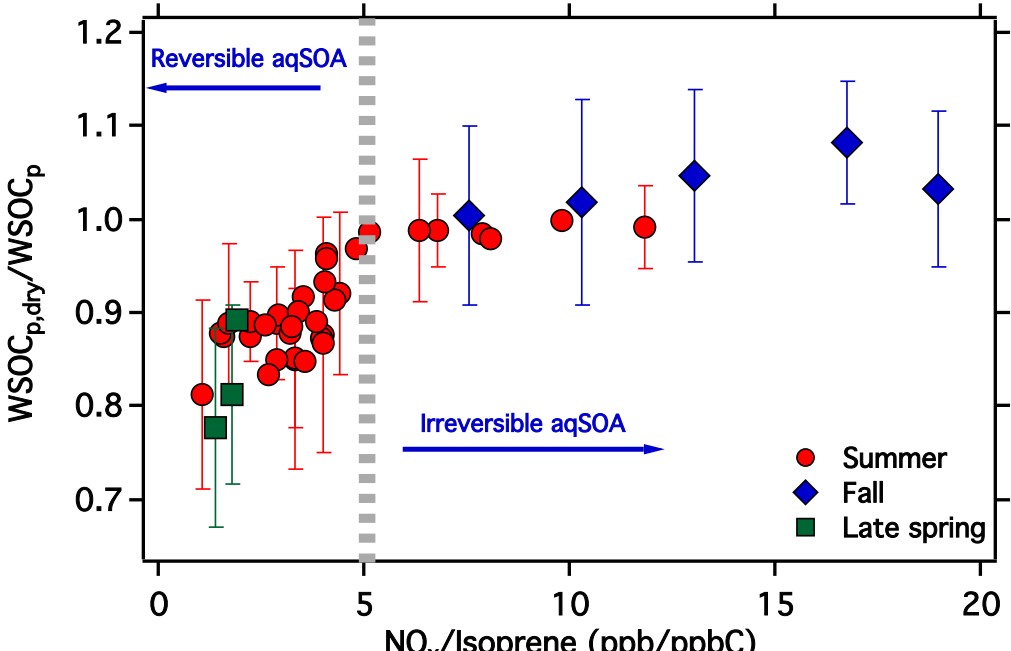

**Figure 7:** Daily average $WSOC_{p,dry}/WSOC_p$ ratios as a function of daily average $NO_x$/isoprene ratios. Gray dotted line is representative of the transition zone from reversible to irreversible aqSOA conditions. Error bars, based on mean $\pm 1\sigma$, are shown for one-third of the data for clarity.