# Peer review of "The effects of isoprene and NOx on secondary organic aerosols"

_Atmospheric Chemistry and Physics, 2017_

## Referee Comment (RC1) · Anonymous Referee #1 · 13 Sep 2017

Summary:

This article seeks to assess the reversibility of aqueous secondary organic aerosol (aq-SOA). The methodology, in this work, is implemented to sample tropospheric aerosols and probe the aqSOA contents within. The authors infer that the aqSOA is primarily isoprene-derived, and attempt to elucidate the influence of NO$_X$ on the extent of reversibility. They use a Particle-Into-Liquid-Sampler (PILS) coupled to a Total Organic Carbon (TOC) analyzer to measure aqSOA / water soluble organic carbon (WSOC) content, with a custom-made mist chamber and denuders as conditioning apparatus prior to sampling. The gas-phase measurements however were not conducted by the

authors, rather they were obtained by the Maryland Department of the Environment (MDE) located ∼20 km from their sampling site. Taken together, the results from the PILS/TOC and gases (isoprene and NOX) seem to suggest that low-NOX isoprene-derived aqSOA is more prone to reversibility than high-NOX isoprene-derived aqSOA. The literature does not seem to be abundant enough – in context of reversibility – to compare to the measurements, making this study unique.

Perhaps the most interesting segment of this article is the time-lag analysis that correlates isoprene to water-soluble organic carbon (WSOC), acting as a proxy that crudely considers transport of isoprene-laden air from the source to the sampling site. Some seasonal analysis is done that suggests both secondary organic aerosol (SOA) and aqSOA abundance is correlated to summertime isoprene mixing ratios, further suggesting the reversibility of aqSOA is driven by isoprene oxidation products. That said, no back trajectories are included in the article. If the authors are correct, accounting for reversibility of aqSOA (or SOA in general) can non-negligibly influence aerosol loadings in certain continental areas.

Overall, this article presents an interesting study and tackles an important area of aerosol chemistry and isoprene chemistry. However, in my view, it is not clearly written. Concepts do not come across easily, neither in explanations nor in inferences. While the science is appropriate for ACP and an ACP audience, the analysis and language need to be cleaned up. I recommend this be published in ACP once my comments are addressed, as it can lay groundwork for more studies of its kind.

Major comments:

While the authors demonstrate there is a relationship between isoprene and aqSOA (or WSOC, depending on the definition) reversibility, implying isoprene-derived aqSOA is at least ∼25% reversible, their data analysis could be a lot stronger. Several figures (3-6) don't have error bars nor do they include the full data, e.g. scattered behind the trends. Because this is not a modeling paper, rather a purely experimental one,

rigorous data analysis needs to be included for ACP standards.

Furthermore, the Atmospheric Implications section and any discussion that follows lacks some key components. For example, peroxymethacryloyl nitrate (MPAN) is a known NOX reservoir formed through the photooxidation of biogenic hydrocarbons (Bertman and Roberts, 1991; Tuazon and Atkinson, 1990), yet it is not mentioned in any high-NOX scenarios. Perhaps it would be worthwhile to include some mention of MPAN and how it can affect aqSOA reversibility. Have any studies of MPAN formation from aqueous uptake of isoprene been done that can help in this discussion (Surratt et al., 2009)? Without this, the discussion of NOX influence on aqSOA appears shallow. Several times throughout the document the authors specify that low-NOX conditions are responsible for reversible aqSOA yet the only compounds mentioned are isoprene epoxydiol (IEPOX), glyoxal, methylglyoxal, and other low-NOX products. With the increase of anthropogenic activity, this may warrant further discussion.

With regards to timeseries, I wonder why the authors do not include them anywhere (except for isoprene). In the Supplement, there is a diurnal (diel) profile that suggests data was taken, or averaged, every hour, at least during the summertime. It would be great to have a timeseries for the year of isoprene, NOX, and WSOC so that the data in this manuscript can come into context, e.g. Fig. 1. This timeseries can fit in the Supplement in my opinion. In the same vein, Fig. S2 could come with confidence intervals, and perhaps Fig. 1 could have 12 box-and-whiskers (one for every month) to better capture seasonal variability. If data is insufficient, the authors should place more effort in explaining that.

In addition, to bolster time lag arguments and correlations, if windroses are not available from MDE then perhaps some back trajectories can be calculated to ensure time lag air masses do not mix, e.g., with other air masses, the free troposphere, etc. While at the beginning of Section 3.3 the authors provide a brief discussion on atmospheric lifetimes, that can be expanded with the inclusion of transport. Further literature reading is encouraged on that front.

Page 9 Line 8: In my opinion, this paragraph should be moved to the beginning of the results section! I found it to be a great paragraph. Readers may be confused as to why the authors don't explain what the results really mean – which if I understand correctly is that IEPOX reversibly partitions – until after a discussion of how aqSOA reversibility can affect model predictions! I felt as though I kept guessing what their results meant and why the authors chose this method of drying coupled to a mist chamber.

The Uncertainties section label may be misconstrued. There are no quantitative arguments in the section, let alone statistical error analyses, just qualitative interpretations of the data obtained. I would revise the section caption or move the text to a different section or sub-section.

In the Conclusion section, the first paragraph reads: "Lower NOX leads to increase SOA production..." This needs to be revisited. It is believed (Spracklen et al., 2011), as the Southern Oxidant and Aerosols Study (SOAS) campaign also suggest, that higher NOX mixing ratios enhance SOA production. If the authors are talking specifically about reversible aqSOA, they need to state that clearly, and that otherwise their surrogate is not representative of (urban) continental SOA.

A schematic / diagram of the setup is highly encouraged. This would help envision the split of WSOCp and WSOCg.

For my clarification, can the authors explicitly state the difference between aqSOA and WSOCp? I'm assuming a major difference is that WSOCp can be primary organic aerosol (POA), but the audience may miss this. Also for my clarification, does 're-versible' imply physical partitioning or chemical equilibria? Or both?

Finally, I think the Supplement should at least contain the title and author list.

Minor Comments:

Page 1 Line 27: "The oxidation of isoprene has important implications..." – consider revising or removing 'important implications' redundancy and nuancing how isoprene

oxidation results in SOA, e.g.: "Isoprene oxidation is known to stimulate tropospheric O3 production and contributes to SOA formation, thus affecting the local environment". Relevant literature should be cited, e.g (Claeys, 2004; Kamens et al., 1982; Kroll et al., 2006).

Page 1 Line 28: "In regions with high isoprene emissions, such as the southeastern United States, isoprene is..." – perhaps consider revising sentence structure to avoid repeating the word 'isoprene' twice in a sentence. Furthermore, citing two articles that don't conclude isoprene by itself is the major SOA precursor can be scant. While the Ozarks are known as the 'isoprene volcano', other terpenes (with SOA yields much higher than isoprene) can compete for total SOA load. If the authors can either rephrase the sentence to imply that isoprene is an important SOA precursor versus 'the' dominant SOA precursor, the sentence can be justified by citing the two articles.

Page 1 Line 31: "...glyoxal and methylglyoxal." – consider an Oxford comma unless aldehydes are meant to be lumped together as a class separate from epoxides.

Page 2 Line 1: "A body of work indicates..." – while studies suggest uptake of organic gases in water lead to brown carbon formation, it should be pointed out that photochemical SOA production from isoprene occurs during homogeneous and heterogeneous nucleation (chamber studies), implying aqueous uptake is not the only source of isoprene SOA. A clarification is encouraged.

Page 2 Line 20: Consider replacing the semicolon by a full stop to break the sentence.

Page 2 Line 34: I would think this sentence is better fit at the end of the previous paragraph.

Page 2 Line 37: Consider substituting 'reaction' with 'oxidation'.

Page 2 Line 37: "This includes a major effect on isoprene oxidation chemistry, ..." what does that mean? Is the major effect simply high and low yield? Or is it differences in chemical pathways? Also consider expanding the literature cited.

Page 3 Line 5: Consider rewording "with our understanding" to "with the understanding".

Page 3 Line 9: Consider citing more literature, e.g. (Kroll et al., 2006; Lin et al., 2013; Surratt et al., 2006, 2009).

Last paragraph of Introduction: Seems redundant, consider revising or removing.

Page 3 Line 25: Consider using a comma, e.g. "...using a mist chamber (MC), and in the particle phase...". Furthermore, is a brief description of the MC available? For anyone interested in the technique, which may not be as diffuse as the authors imply, it may be cumbersome to backtrack El-Sayed et al. 2015, then Hennigan et al. 2009, then Cofer and Edahl 1986. Diagrams are encouraged.

Page 3 Line 27: Outline the model before explaining what mode it was operated in.

Page 3 Line 28: Why is 'dried' in quotes? Given the brief description and lack of diagram, it can be hard for the reader to put words into context.

Page 3 Line 31: Brand (if any, or if custom made) and dimensions of the parallel plate denuder? What flows can it handle? The gas-phase interferences are not necessarily limited to isoprene oxidation products, is that correct?

Page 4 Line 26: The first paragraph of the Results section... is it common to take measurements so infrequently? What does the literature recommend?

Page 4 Line 29: "...WSOCP measurements has been..." was it one measurement or multiple? Ensure verb matches the subject of the sentence. If plural, then correct to "...WSOCP measurements have been...", whereas if singular, correct to "...WSOCP measurement has been...".

Page 4 Line 31: Consider removing sentence "In this regard...was formed." as it doesn't add critical information sandwiched between two sentences that by themselves give enough information.

Page 4 Line 34: Consider having that formula as an equation with a designated equation number. Also, it appears the subscript 'P' is Italicized outside of the bracket, but not inside, and could be corrected. Also, there appears to be a formatting issue with this paragraph in general.

First two paragraphs of Section 3: Consider merging first two paragraphs in one.

Page 5 Line 5: Sentence starts with "Figure 1 . . .", yet in Line 8 of the same page, sentence starts with "Fig. 1. . .". The authors are invited to check for consistency and formatting guidelines of the journal. This may apply for more than one instance.

Page 5 Line 16: I don't understand the citation to El-Sayed et al., 2016. My understanding is that the values 0.92 and 0.87 for mean WSOCP,dry/WSOCP are from data collected for this manuscript, hence, would not be previously published.

Page 5 Line 19: I don't think this sentence belongs here. Aside from this point being stressed before, it is out of place in this paragraph / section. Statements like these should go at the end of the introduction, and they are already included.

Page 5 Line 34: The authors could take more care with outlining the Aerosol Mass Spectrometer (AMS) rather than introducing an undefined acronym. In that regard, what is an 'IEPOX factor' and how does it relate to source apportionment techniques / AMS?

Page 6 Line 29: The authors suggest their diel profile in Fig. S2 is consistent with their data in Figure 3. I would argue that, 3h lag considered, there ought to be an inflection point during the diurnal morning when as WSOCg increases, isoprene decreases. The authors need to address why that inflection in Fig. S2 is not reflected in Fig. 3, arguably indicating the importance of confidence intervals / error bars during the summertime.

Page 6 Line 29: The authors suggest that the chain of reactions leading isoprene to be converted to WSOCg is ∼3-5h. While the data is convincing, without air mass trajectories or insolation data, incorporated with statistics, this assertion is slightly weak.

[Figure]

Could other VOC or VOC oxidation mechanisms explain WSOC? Is regional terpene, sesquiterpene, or agriculture emission chemistry considered? If it is beyond the scope of the article it should be stated.

Page 7 Lines 13-14: "Consistent with Fig. 3 and Fig. 4..." – I do not understand why WSOCg is strongly correlated with isoprene for lags of 3-5h (Fig. 3) whereas Evaporated WSOCp is correlated with isoprene for lags of 6-11 h? If evaporated WSOCp is an example of reversible aqSOA as is WSOCg by proxy, then if they are produced by the same pathway in the same parcel of air, wouldn't they require the same lag time? If not, and they are two different generation isoprene oxidation products, then why is there a relationship in Fig. S4? This is not clear to me, though perhaps I'm missing something. The following sentence "The above observations suggest that isoprene is strongly linked with the formation of reversible aqSOA in the eastern U.S" therefore does not speak to me.

Page 7 Line 20: A simple phrase at the beginning or end of the sentence explaining why the 9h lag was chosen would be helpful. Even though Fig. 4 can by itself be sufficient for an inference, a verbal explanation is helpful.

Page 7 Line 22: "...it is clear..." – as per my comment on Fig. 5, without box-and-whiskers, the 'dramatic' decrease is not clear. Upon initial inspection, it would appear most of the data does not exceed 1 ug/m3, thus invalidating the 'dramatic' decrease.

Page 7 Line 34: Consider rephrasing.

Page 8 Line 15: Awkward phrase: "These results represent, to our knowledge, the first observations to characterize the seasonal occurrence of..." consider revising to, e.g., "To the best of our knowledge, observations of seasonal dependence of reversible aqSOA are reported for the first time in this work.".

Page 8 Line 16: "important implications" has been used 2 out of 3 times in this document at this point. I wonder if it becomes a redundancy. Consider substituting with,

e.g., "affect measurement techniques" or something less vague.

Page 8 Line 21: Consider removing "...to confirm this hypothesis."

Page 8 Line 22: I don't believe the acronym 'AOD' has been defined before by the authors.

Page 9 Line 7: The last sentence is very vague by itself. The paragraph, in general, appears out of place. It is a good point by the authors, but does not seem fit between discussion of aqSOA reversibility on model prediction and discussion of their observations; rather, it can be moved to the end as an anecdotal sentence, or, if elaborated, a paragraph on its own.

Page 10 Line 14: If the effect of ALW is more pronounced at low organic concentrations, why is there no discussion about salting out effects, Raoult's law, etc.?

Page 10 Line 16: "Our observations show..." – if the authors cite their previous publication, I would recommend revising the sentence to "Previous results from our group show..." or words to that effect.

Page 10 Line 18: The authors have not defined neither LVOOA nor SVOOA before, unless I missed it.

Page 10 Line 23: Consider an Oxford comma.

Page 10 Line 25: "They dealt with this problem by..." sounds too colloquial. Consider revising.

Page 11 Line 17: Remove first sentence.

Comments on Figures and Tables

Table S1: Along the same lines of my comments for Page 3 Line 28, this table is not very helpful. It takes a while to understand it. Are the standard deviations for the duration of the study? How often were these measurements made? Would a

timeseries help? Why was the diffusion drier not sized to handle a 90% RH stream and reducing it to <20% RH? What were the dimensions? These details could go in the Supplement (in my opinion).

Figure 1: With the understanding that the authors composed a box-and-whiskers diagram to visualize their data, can something be done about the x-axis potentially misleading a reader that all five data are not evenly spaced across the year? If not, that is OK in my view, but if the data can be displayed with the x-axis being more akin to Date-Time, it would better visualize (in my opinion) the seasonal cycles the authors wish to present.

Figure 2: Upon reading the caption, this is an annual profile averaged across 5 years. I would request the data be replotted using markers and lines, at least, and ideally with some form of confidence intervals to reflect the averaged data. While the point of the authors is that isoprene is high during the summer months, the data can be presented with a little more rigor and care. If data from MDE comes like this, the authors can state it.

Figure 3: If the authors claim that their calculation (or rather, literature review) of isoprene lifetime to OH oxidation is on the order of 1-2h, then this figure really requires at least vertical error bars. While the median WSOCg does correlate with isoprene mixing ratios at lag times between 3-5 h, other types of statistics are encouraged for the argument to be valid.

Figure 5: Consider visuals, at least on the x-axis, to show regime of polluted vs clean air (low values on the x-axis are clean; high values are polluted). Also, if formatting permits, vertical box plots could help visualize the binning. In my opinion, the graph is very misleading otherwise.

References

Bertman, S. B. and Roberts, J. M.: A PAN analog from isoprene photooxidation, Geo-

phys. Res. Lett., 18(8), 1461–1464, doi:10.1029/91GL01852, 1991.

Claeys, M.: Formation of Secondary Organic Aerosols Through Photooxidation of Isoprene, Science, 303(5661), 1173–1176, doi:10.1126/science.1092805, 2004.

Kamens, R. M., Gery, M. W., Jeffries, H. E., Jackson, M. and Cole, E. I.: Ozone-isoprene reactions: Product formation and aerosol potential, Int. J. Chem. Kinet., 14(9), 955–975, doi:10.1002/kin.550140902, 1982.

Kroll, J. H., Ng, N. L., Murphy, S. M., Flagan, R. C. and Seinfeld, J. H.: Secondary Organic Aerosol Formation from Isoprene Photooxidation, Environ. Sci. Technol., 40(6), 1869–1877, doi:10.1021/es0524301, 2006.

Spracklen, D. V., Jimenez, J. L., Carslaw, K. S., Worsnop, D. R., Evans, M. J., Mann, G. W., Zhang, Q., Canagaratna, M. R., Allan, J., Coe, H., McFiggans, G., Rap, A. and Forster, P.: Aerosol mass spectrometer constraint on the global secondary organic aerosol budget, Atmospheric Chem. Phys., 11(23), 12109–12136, doi:10.5194/acp-11-12109-2011, 2011.

Surratt, J. D., Murphy, S. M., Kroll, J. H., Ng, N. L., Hildebrandt, L., Sorooshian, A., Szmigielski, R., Vermeylen, R., Maenhaut, W., Claeys, M., Flagan, R. C. and Seinfeld, J. H.: Chemical Composition of Secondary Organic Aerosol Formed from the Photooxidation of Isoprene, J. Phys. Chem. A, 110(31), 9665–9690, doi:10.1021/jp061734m, 2006.

Surratt, J. D., Chan, A. W. H., Eddingsaas, N. C., Chan, M., Loza, C. L., Kwan, A. J., Hersey, S. P., Flagan, R. C., Wennberg, P. O. and Seinfeld, J. H.: Reactive intermediates revealed in secondary organic aerosol formation from isoprene, Proc. Natl. Acad. Sci., 107(15), 6640–6645, doi:10.1073/pnas.0911114107, 2009.

Tuazon, E. C. and Atkinson, R.: A product study of the gas-phase reaction of Methacrolein with the OH radical in the presence of NOx, Int. J. Chem. Kinet., 22(6), 591–602, doi:10.1002/kin.550220604, 1990.

---

## Referee Comment (RC2) · Anonymous Referee #2 · 18 Sep 2017

General comments: This work examines aqueous SOA, both reversible (able to evaporate upon drying) and irreversible, in the Eastern US using measurements of water soluble compounds in both the gas and particle phase. Additional measurements (isoprene, NOx) are used to infer that the reversible SOA is a result of isoprene epoxydiol (IEPOX) uptake to an aqueous medium. This paper examines an important issue with implications for what controls IEPOX SOA formation. However, to further their conclusions it would be good to demonstrate stronger connections between isoprene and the reversible SOA since no chemical identity beyond WSOCp (particulate WSOC) and WSOCp,dry (dried WSOC) is known for the organic aerosol. The major pieces of evidence for IEPOX being the precursor to reversible SOA come from NOx and isoprene

concentrations and time lag analysis. The WSOCp peaks 9 hours after isoprene (consistent with IEPOX being 2nd generation plus an additional lag), and the reversible SOA is highest when NOx/Isoprene is lowest which is consistent with our understanding of IEPOX formation in the gas-phase. However, formation of IEPOX may not be the limiting factor for IEPOX SOA formation (sulfate and its influence on particle surface area/volume as well as acidity may be responsible). In addition, other aspects of the ambient atmosphere are changing in addition to NOx and isoprene as a function of season. Two areas that could be furthered include:

1. Can mass closure be reached in terms of how much isoprene is present and the amount of WSOCp and WSOCg?

a. E.g. Page 6 line 25: Do you get mass closure if you assume 5 ppbC of isoprene reacted forms 2 ugC/m3 IEPOX?

b. Is the reversible IEPOX SOA just dissolved IEPOX or is it a reversibly formed reaction product? Are the levels of reversible IEPOX SOA consistent with dissolved IEPOX? Sareen et al. 2017 indicate dissolved IEPOX alone is a very small concentration (especially compared to IEPOX SOA from AMS PMF analysis).

2. Were other proxies for chemistry besides NOx/Isoprene examined?

a. Page 7, near line 10: Is the diurnal variation in sulfate involved in IEPOX SOA?

b. Figure 7 shows seasonality in the WSOCp,dry/WSOCp ratio consistent with changes in NOx/Isoprene. What else changes as a function of season that could also explain the ratio? Oxidants? How is ALW changing? If the horizontal axis was sulfate or SOx divided by isoprene would it show the same behavior?

Other Specific Comments:

3. Page 1: Lines 23-24 indicate that the trend towards lower NOx/Isoprene ratios may mean more IEPOX SOA in the future. Given the dependence of IEPOX SOA on sulfate, wouldn't we expect this pathway to decrease with decreasing sulfate levels in the future as demonstrated by Marais et al. 2017 ERL (http://iopscience.iop.org/article/10.1088/1748-9326/aa69c8/meta)?

4. Page 1: Line 29 indicates isoprene is the dominant SOA precursor in summer. I would define dominant as responsible for >= 50% of SOA. Hu et al. 2015 ACP (https://doi.org/10.5194/acp-15-11807-2015) indicate isoprene (or IEPOX) is responsible for 17% to 36% of Southeast US SOA. So while it is important, it is not dominant.

5. Page 2: Lines 21-23: I would characterize both Marais et al. 2016 and Pye et al. 2013 as irreversible IEPOX uptake since both use a reactive uptake formulation. The major difference between Marais et al. and Pye et al. is the Henry's law coefficient which leads to different amounts of IEPOX SOA. They also simulated different years. Budisulistiorini et al. 2017 has shown that reversible (simpleGAMMA, McNeill et al. 2012) and irreversible (CMAQ, Pye et al. 2013) models of IEPOX uptake can agree when the parameters going into them are identical (for ∼6 hours of processing time).

6. Page 3: Line 17-18: which instruments may not measure reversible SOA?

7. Page 4: Near line 30: Can you clarify the relationship between WSOCp and aqSOA? What fraction of WSOCp is aqSOA? How was aqSOA identified?

8. Page 8: Line 35: How much higher is the fraction of reversible aqSOA? Insert value

9. Page 8: Line 36: For the range of 0-60%, what is the typical value (mean, median, or similar)?

10. Page 9: Line 18-22: I am unclear as to whether or not the work of Wong et al., 2015 is atmospherically relevant if their experiments did not produce SOA from IEPOX. D'Ambro et al. 2017 ES&T (http://pubs.acs.org/doi/abs/10.1021/acs.est.7b00460) demonstrates that IEPOX is the atmospherically relevant pathway to isoprene SOA and laboratory experiments with unrealistic concentrations may be activating pathways that are not important in the atmosphere.

---

## Referee Comment (RC3) · Anonymous Referee #3 · 19 Sep 2017

Summary

This article examines the influence of NOx on the reversibility of aqueous SOA. The paper analyzed the irreversible and reversible water-soluble organic carbon (WSOC) in the particle phase, as well as WSOC in the gas phase from a site in the Eastern US. By correlating the gas phase and particle phase WSOC with isoprene concentration measured at a nearby site (∼20 km away), the author believes that IEPOX uptake is responsible for the reversible aqueous SOA. By correlating the NOx/isoprene ratio vs. the percentage of reversible aqueous SOA of total particle phase WSOC, the author suggests that low NOx/isoprene ratios seem to enhance reversible aqueous SOA for-

mation, which agrees with the author's assumption that IEPOX is the main reason for reversible aqSOA formed in late spring and summer.

This study provides a nice perspective of how reversible WSOC could affect the SOA budget and how drying the aerosol before performing field measurement could neglect reversible WSOC. The work performed to attribute the sources of such reversible WSOC by performing a time lag correlation on the isoprene concentration is interesting, but the lack of further analysis on the molecular composition of the gas and particle phase WSOC weakens the conclusion.

Overall, this article provides a unique perspective on the importance of WSOC in SOA with enough scientific content and novelty to be published in ACP. However, the conclusion that IEPOX is the likely cause of reversible WSOC in aqSOA is not strong enough and the logic between sentences in some paragraphs is not clear. The author needs to address the following issues and refine the wording before being published in ACP.

Major Comments

The sampling site of WSOC (Baltimore) is ~20 km away from the site sampling isoprene and NOx (Essex site), so whether the Essex site can be representative of the Baltimore site is a questionable part of this study, especially when the Baltimore site is heavily influenced by anthropogenic emissions and the author showed up to 11 hour lag comparison between the two sites. Back trajectory data would be better to use in this paper to justify the result, in order to (1) either prove that Baltimore is downwind of the Essex Site, (2) or to filter out those data when Baltimore is not downwind of the Essex site.

It is difficult to make a strong argument that IEPOX is the main reason for the reversible WSOC in the particle phase without chemical characterization. Other BVOCs (such as monoterpenes) can also form water soluble components that were shown to enhance SOA mass at high RH (Prisle et al., 2010), and their reaction mechanisms are also sensitive to NOx concentration (Wildt et al.). Even though IEPOX may seem to be a

more likely compound for reversible aqSOA for this study, the author needs to provide stronger evidence to rule out other possibilities, such as a correlation plot between particle WSOC vs. time lagged monoterpene concentration.

Page 3, line 7, the author listed the ACP paper by Chan Miller et al. (2017) to show that glyoxal is formed in both low- and high- NOx pathways while IEPOX is mainly formed in the low NOx pathway. Therefore the author states that the correlation between NOx/isoprene can be attributed to IEPOX. But, the yield of glyoxal in high and low NOx conditions are different. Chan Miller et al. shows in his paper (Figure 2) that glyoxal formed from isoprene oxidation has a higher yield at low NOx condition compared with a high NOx condition. Therefore, I believe the increase of reversible aqSOA could be at least partially attributed to glyoxal. The author needs to specify all these possibilities in the paper rather than attributing the reversible aqSOA solely on IEPOX. The conclusion in the abstract as well as throughout the paper is too strong and needs to be revised.

Besides comparing WSOCp, dry/WSCOp with isoprene concentration, has the author studied the influence of ambient humidity on WSOCp, dry/WSCOp? Moreover, from Table S1, it seems that when ambient RH=80%, the RHs of the samples passed through the silica gel dryer were consistently higher in the summer time than in the winter time. What are the reasons and would that cause artifacts of the results?

Because IEPOX usually undergoes reactive uptake with high acidity aerosols (Gaston et al., 2014; Riedel et al., 2015), people have been assuming that isoprene-derived SOA is not very important in low acidity aerosols. However, the result presented in this study shows the importance of isoprene-derived SOA even for low acidity aerosol particles, especially when there is an amount of liquid water in the aerosol so IEPOX can have reversible partitioning. The author should probably talk about the importance of this aspect in the atmospheric implication section as well.

Minor Comments

Page 1, line 31. Oxford comma is recommended here after glyoxal.

Page 2, line 1. Besides all the literature the author listed here, I believe Riedel et al., (ES&T Letters, 2015) should also be included as well when talking about reactive uptake of IEPOX.

Page 2, line 14. The author used an incorrect example here. Oligomerization is a non-reversible process, as also shown in De Haan et al. that the author cited.

Page 2, line 15. Oxford comma is recommended here after inorganics.

Page 3, line 25. Because the author performed the experiment using a home-built mist chamber, is there any characterization of this mist chamber, such as recovery efficiency of the gas phase species? Such information would help the reader in understanding the performance of the mist chamber and also error bar of the measurement.

Page 5, line 10. Has the author compared the ambient RH of summer and winter times? Does ambient RH have an effect on WSOCp, dry/WSCOp

Page 5, line 33-34. As previously mentioned, it would be better if the author could compare other BVOCs with WSOC obtained from this study to rule out the possibilities of other BVOCs producing WSOC.

Page 7, line 21. Has the author examined the relationship between ozone and WSOCp? If there is a correlation, then it means other BVOCs can also contribute to WSOCp as well.

Page 7, line 33-35. The sentence "If isoprene is indeed. . ." seems to be out of the place here because it does not go with the sentence below logically. The author can either elaborate more on this sentence or delete this sentence.

Page 8, line 22. AOD was not defined previously. Please define.

Page 8, line 23-line 36. This paragraph is pretty confusing because there are different concepts and ideas intertwined with each other. The author can talk about the results in Fig. 1 first, and then mention Liu et al. and Riva et al., and lastly talk about McNeil

et al.

Page 8, line 32-line 33. What are the traditional SOA pathways? I would recommend specifying it more clearly because "uptake to aqueous particles" sounds like a traditional pathway of SOA formation to me as well.

Page 8, line 36. Perhaps I am missing something here. Why is the reversible aqSOA even higher given that ~10 to 15% of the total WSOCp evaporates?

Page 9, line 15. The reason that ambient IEPOX-SOA has a low volatility can additionally be attributed to the higher viscosity of SOA. If the viscosity of the SOA is higher, then it will be more difficult for the semi-volatile species to evaporate and escape from the particle phase within the timescale of the measurement, as discussed in Vaden et al. (2011) and Zhang et al. (2015). This can be another reason why some ambient aerosols do not show strong reversibility and the author should consider putting it in the discussion.

Page 10, line 13-line 21. This paragraph is confusing as well. At the beginning of the paragraph the author seems to believe the effect of ALW on WSOC is not as significant as OA concentration. By the end of the paragraph the author concludes that WSOC is not due to OA partitioning. Please revise this paragraph to give a clearer explanation.

Page 10, line 18. Please define LV-OOA and SV-OOA before using these two terms.

References

Prisle, N. L., Engelhart, G. J., Bilde, M. & Donahue, N. M. Humidity influence on gas-particle phase partitioning of $\alpha$-pinene + O3 secondary organic aerosol. Geophys Res Lett. 37 (1), doi:10.1029/2009GL041402, (2010).

Wildt, J. et al. Suppression of new particle formation from monoterpene oxidation by NOx. Atmos Chem Phys. 14 (6), 2789-2804, doi:10.5194/acp-14-2789-2014, (2014).

Chan Miller, C. et al. Glyoxal yield from isoprene oxidation and relation to formaldehyde: chemical mechanism, constraints from SENEX aircraft observations, and interpretation of OMI satellite data. Atmos Chem Phys. 17 (14), 8725-8738, doi:10.5194/acp-17-8725-2017, (2017).

Riedel, T. P. et al. Heterogeneous Reactions of Isoprene-Derived Epoxides: Reaction Probabilities and Molar Secondary Organic Aerosol Yield Estimates. Environmental Science & Technology Letters. 2 (2), 38-42, doi:10.1021/ez500406f, (2015).

Vaden, T. D., Imre, D., Beránek, J., Shrivastava, M. & Zelenyuk, A. Evaporation kinetics and phase of laboratory and ambient secondary organic aerosol. Proc Natl Acad Sci USA. 108 (6), 2190-2195, doi:10.1073/pnas.1013391108, (2011).

Zhang, Y. et al. Changing shapes and implied viscosities of suspended submicron particles. Atmos Chem Phys. 15 (14), 7819-7829, doi:10.5194/acp-15-7819-2015, (2015).

A formatted version of the above comments is attached in the supplement.

Please also note the supplement to this comment:
https://www.atmos-chem-phys-discuss.net/acp-2017-702/acp-2017-702-RC3-supplement.pdf

---

## Author Comment (AC1) · 1 Dec 2017

**Response to Reviews**

We thank the reviewers for their detailed comments and helpful suggestions. We have addressed each comment below, with the Referee comment in bold italicized text, our response in plain text, and any manuscript changes noted in red text. In addition, the revised manuscript with changes marked up has been attached to the end of our response to Referee 3.

Anonymous Referee #1

*This article seeks to assess the reversibility of aqueous secondary organic aerosol (aqSOA). The methodology, in this work, is implemented to sample tropospheric aerosols and probe the aqSOA contents within. The authors infer that the aqSOA is primarily isoprene-derived, and attempt to elucidate the influence of $NO_x$ on the extent of reversibility. They use a Particle-Into-Liquid-Sampler (PILS) coupled to a Total Organic Carbon (TOC) analyzer to measure aqSOA / water soluble organic carbon (WSOC) content, with a custom-made mist chamber and denuders as conditioning apparatus prior to sampling. The gas-phase measurements however were not conducted by the authors, rather they were obtained by the Maryland Department of the Environment (MDE) located ~20 km from their sampling site. Taken together, the results from the PILS/TOC and gases (isoprene and $NO_x$) seem to suggest that low- $NO_x$ isoprene derived aqSOA is more prone to reversibility than high- $NO_x$ isoprene-derived aqSOA. The literature does not seem to be abundant enough – in context of reversibility – to compare to the measurements, making this study unique. Perhaps the most interesting segment of this article is the time-lag analysis that correlates isoprene to water-soluble organic carbon (WSOC), acting as a proxy that crudely considers transport of isoprene-laden air from the source to the sampling site. Some seasonal analysis is done that suggests both secondary organic aerosol (SOA) and aqSOA abundance is correlated to summertime isoprene mixing ratios, further suggesting the reversibility of aqSOA is driven by isoprene oxidation products. That said, no back trajectories are included in the article. If the authors are correct, accounting for reversibility of aqSOA (or SOA in general) can non-negligibly influence aerosol loadings in certain continental areas. Overall, this article presents an interesting study and tackles an important area of aerosol chemistry and isoprene chemistry. However, in my view, it is not clearly written. Concepts do not come across easily, neither in explanations nor in inferences. While the science is appropriate for ACP and an ACP audience, the analysis and language need to be cleaned up. I recommend this be published in ACP once my comments are addressed, as it can lay groundwork for more studies of its kind.*

**Major comments:**

*1. While the authors demonstrate there is a relationship between isoprene and aqSOA (or WSOC, depending on the definition) reversibility, implying isoprene-derived aqSOA is at least ~25% reversible, their data analysis could be a lot stronger. Several figures (3-6) don't have error bars nor do they include the full data, e.g. scattered behind the trends. Because this is not a modeling paper, rather a purely experimental one, rigorous data analysis needs to be included for ACP standards.*

We have updated Figures 2, 3, 4, 5, and 6 in response to the Referee's comment (see also our response to comments #55 through #59 below).

***2. Furthermore, the Atmospheric Implications section and any discussion that follows lacks some key components. For example, peroxymethacryloyl nitrate (MPAN) is a known $NO_x$ reservoir formed through the photooxidation of biogenic hydrocarbons (Bertman and Roberts, 1991; Tuazon and Atkinson, 1990), yet it is not mentioned in any high-$NO_x$ scenarios. Perhaps it would be worthwhile to include some mention of MPAN and how it can affect aqSOA reversibility. Have any studies of MPAN formation from aqueous uptake of isoprene been done that can help in this discussion (Surratt et al., 2009)? Without this, the discussion of $NO_x$ influence on aqSOA appears shallow.***

From Sander (2015), the Henry's law constant for MPAN is quite low (1.7 M atm$^{-1}$), ≈five orders of magnitude (at least) lower than the Henry's law constant for glyoxal. Further, Pye et al. (2017) suggest that organic nitrates are among the least soluble SOA species. Therefore, we do not anticipate a significant contribution of MPAN uptake to form aqSOA.

***3. Several times throughout the document the authors specify that low-$NO_x$ conditions are responsible for reversible aqSOA yet the only compounds mentioned are isoprene epoxydiol (IEPOX), glyoxal, methylglyoxal, and other low- $NO_x$ products. With the increase of anthropogenic activity, this may warrant further discussion.***

We are unclear what the Referee is specifically referring to with "the increase of anthropogenic activity"? Most relevant to the context of this study, $NO_x$ concentrations in the eastern U.S. are declining, and we note in the manuscript that isoprene oxidation in this region occurs approximately equally between the high- and low-$NO_x$ pathways (citing Travis et al. (2016)).

Is the Referee instead suggesting that we be more specific instead of using "other low-$NO_x$ products"? This will be challenging with the current body of literature. Surratt et al. (2010) show that ISOPOOH does not undergo uptake to aqueous particles (acidic or neutral). Wong et al. (2015) show reversible aqSOA from non-IEPOX products of low-$NO_x$ isoprene oxidation, but they do not provide molecular identification of the gaseous precursors. Studies also show non-IEPOX isoprene SOA (e.g., (Liu et al., 2016)), but under lab conditions not relevant to the eastern U.S. (without acidic aqueous seed particles). In summary, there is simply not enough known about the molecular identities of other low-$NO_x$ isoprene oxidation products that may form aqSOA.

***4. With regards to timeseries, I wonder why the authors do not include them anywhere (except for isoprene). In the Supplement, there is a diurnal (diel) profile that suggests data was taken, or averaged, every hour, at least during the summertime. It would be great to have a timeseries for the year of isoprene, $NO_x$, and WSOC so that the data in this manuscript can come into context, e.g. Fig. 1. This time series can fit in the Supplement in my opinion. In the same vein,***

*Fig. S2 could come with confidence intervals, and perhaps Fig. 1 could have 12 box-and-whiskers (one for every month) to better capture seasonal variability. If data is insufficient, the authors should place more effort in explaining that.*

Figure 2 presents a climatology of isoprene concentrations averaged over five years. Our WSOC measurements were not carried out for a year continuously, so a time series of these species would look very different than that of Figure 2 (with a lot of empty space). Table 1 in the manuscript clearly identifies the dates that correspond to the sampling within each season. We have added standard deviations to Figure 2, and Figure S2, and have added to the Supplement box plots showing individual data points (and their statistics) corresponding to Figures 3 and 4. We have also added box plots on top of the individual data plotted in Figures 5 and 6.

*5. In addition, to bolster time lag arguments and correlations, if windroses are not available from MDE then perhaps some back trajectories can be calculated to ensure time lag air masses do not mix, e.g., with other air masses, the free troposphere, etc.*

In this case, we do not agree that a back trajectory analysis is required to support our correlations. We have added the following discussion to Section 2 to better explain and justify our methods: "A key assumption employed in this analysis is that the WSOC measurements made at UMBC are representative of conditions at Essex, the location of the $NO_x$ and isoprene measurements. Aerosol concentrations in the Baltimore-Washington region are spatially uniform over tens of kilometers (Beyersdorf et al., 2016). Further, $WSOC_p$ concentrations exhibit small spatial variations across urban-to-rural gradients during the summertime (Weber et al., 2007). These prior analyses showed that aerosol concentrations, and in particular WSOC, were not dependent on wind direction. Isoprene emissions in the eastern U.S. are regional in nature, due to the expansive coverage of broadleaf forests (Guenther et al., 2012; Pye et al., 2013). $NO_x$ emissions are spatially segregated from those of isoprene, and are far more localized. However, the isoprene-$NO_x$ chemical regime (high- or low-$NO_x$) in the eastern U.S. is generally well-represented with model resolution of 28 x 28 km, suggesting that the chemistry occurring on small scales, such as in individual power plant plumes, does not significantly affect the regional isoprene-$NO_x$ regime (Yu et al., 2016). $NO_x$ concentrations at Essex (20 km ENE of UMBC) and HU-Beltsville (35 km SSW of UMBC) are strongly correlated (R = 0.89, Fig. S6), likely due to the overwhelming contribution of mobile source emissions along the heavily-traveled I-95 corridor to the region (Anderson et al., 2014). Together, this supports our analysis into the effects of isoprene and $NO_x$ on reversible aqSOA using the measurements described above."

*6. While at the beginning of Section 3.3 the authors provide a brief discussion on atmospheric lifetimes, that can be expanded with the inclusion of transport. Further literature reading is encouraged on that front.*

We point the Referee to our discussion in Section 4, Atmospheric Implications:
"The lifetime of organic compounds in the atmosphere is strongly dependent on their phase (Pye et al., 2017). Oxygenated organic compounds in the gas-phase often have much shorter lifetimes

than particle-phase organics due to significantly higher dry deposition velocities (Nguyen et al., 2015) and photolysis rates (Fu et al., 2008). Thus, the reversible uptake of $WSOC_g$ to aerosol water may effectively shield these species from such loss processes, resulting in enhanced transport. Accounting for the reversible partitioning of water-soluble organic gases to aerosol water would likely improve model predictions of these compounds."

*7. Page 9 Line 8: In my opinion, this paragraph should be moved to the beginning of the results section! I found it to be a great paragraph. Readers may be confused as to why the authors don't explain what the results really mean – which if I understand correctly is that IEPOX reversibly partitions – until after a discussion of how aqSOA reversibility can affect model predictions! I felt as though I kept guessing what their results meant and why the authors chose this method of drying coupled to a mist chamber.*

We agree with the Referee's suggestion and have moved this paragraph to the beginning of Section 4.

*8. The Uncertainties section label may be misconstrued. There are no quantitative arguments in the section, let alone statistical error analyses, just qualitative interpretations of the data obtained. I would revise the section caption or move the text to a different section or sub-section.*

We agree with the Referee's comment. We have removed this section label, and have moved each of the paragraphs to their most relevant section.

*9. In the Conclusion section, the first paragraph reads: "Lower $NO_x$ leads to increase SOA production. . ." This needs to be revisited. It is believed (Spracklen et al., 2011), as the Southern Oxidant and Aerosols Study (SOAS) campaign also suggest, that higher $NO_x$ mixing ratios enhance SOA production. If the authors are talking specifically about reversible aqSOA, they need to state that clearly, and that otherwise their surrogate is not representative of (urban) continental SOA.*

The Referee is correct – we have clarified the sentence so that it now reads: "Lower $NO_x$ leads to a higher fraction of aqueous SOA formed reversibly."

*10. A schematic / diagram of the setup is highly encouraged. This would help envision the split of WSOCp and WSOCg.*

In two prior papers from our group (El-Sayed et al., 2016; 2015), both of which are cited here, we have included a schematic of the experimental setup. See our response to comment #55 below, as well.

*11.  For my clarification, can the authors explicitly state the difference between aqSOA and WSOCp? I'm assuming a major difference is that WSOCp can be primary organic aerosol (POA), but the audience may miss this. Also for my clarification, does 'reversible' imply physical partitioning or chemical equilibria? Or both?*

We have added the following to the Methods section: "$WSOC_p$ is operationally defined based upon the solubilites of the organics, themselves, and the level of dilution employed for the analysis (Psichoudaki and Pandis, 2013).  In the eastern U.S., the $WSOC_p$ measurement is a surrogate for SOA, especially during summer (Weber et al., 2007).  The measurement includes SOA formed through absorptive partitioning and through aqueous-mediated pathways (aqSOA). We consider any $WSOC_p$ that evaporates with drying to be reversible aqSOA, since this material exists in the condensed phase because of the aerosol water and partitions back to the gas phase when the water evaporates."

To the Referee's second point: since the WSOC measurements do not provide molecular information, we do not have the tools to directly characterize the partitioning mechanism. However, we can infer some information about the process, as we have in the Atmospheric Implications section: "Note that Sareen et al. (2017) predict very low dissolved IEPOX in the eastern U.S. during summer ($< 0.01$ µg m$^{-3}$), suggesting reversibly formed reaction products are the dominant contributors to reversible aqSOA."

*12.  Finally, I think the Supplement should at least contain the title and author list.*

We have prepared the Supplement according to the ACP guidelines, which are as follows: "Supplements will receive a title page added during the publication process including title ("Supplement of"), authors, and the correspondence email. Therefore, please avoid providing this information in the supplement."

**Minor Comments:**

*13.  Page 1 Line 27: "The oxidation of isoprene has important implications. . ." – consider revising or removing 'important implications' redundancy and nuancing how isoprene oxidation results in SOA, e.g.: "Isoprene oxidation is known to stimulate tropospheric O3 production and contributes to SOA formation, thus affecting the local environment". Relevant literature should be cited, e.g (Claeys, 2004; Kamens et al., 1982; Kroll et al., 2006).*

We have changed the sentence so that it now reads: "Isoprene oxidation stimulates tropospheric ozone production and contributes substantially to secondary organic aerosol (SOA) formation, thus impacting air quality and climate (Henze and Seinfeld, 2006; Pfister et al., 2008)."

*14.  Page 1 Line 28: "In regions with high isoprene emissions, such as the southeastern United States, isoprene is. . ." – perhaps consider revising sentence structure to*

*avoid repeating the word 'isoprene' twice in a sentence. Furthermore, citing two articles that don't conclude isoprene by itself is the major SOA precursor can be scant.*

We have revised the sentence to: "In the southeastern United States, isoprene is likely the dominant SOA precursor during summer (Kim et al., 2015; Ying et al., 2015)."

**15. While the Ozarks are known as the 'isoprene volcano', other terpenes (with SOA yields much higher than isoprene) can compete for total SOA load. If the authors can either rephrase the sentence to imply that isoprene is an important SOA precursor versus 'the' dominant SOA precursor, the sentence can be justified by citing the two articles.**

The Referee is correct that measurements and models constrain the IEPOX contribution to SOA in the eastern U.S. during summer to less than 50%. However, isoprene also forms SOA that is not necessarily associated with the IEPOX factor identified by the AMS. The studies we have cited here both predict that isoprene is the dominant SOA precursor, contributing more than 50% of SOA in the southeastern U.S during summer. These predictions have uncertainties, though, and to acknowledge this, we have changed the text to read: "In the southeastern United States, isoprene is likely the dominant SOA precursor during summer (Kim et al., 2015; Ying et al., 2015)."

**16. Page 1 Line 31: ". . .glyoxal and methylglyoxal." – consider an Oxford comma unless aldehydes are meant to be lumped together as a class separate from epoxides.**

Comma has been added.

**17. Page 2 Line 1: "A body of work indicates. . ." – while studies suggest uptake of organic gases in water lead to brown carbon formation, it should be pointed out that photochemical SOA production from isoprene occurs during homogeneous and heterogeneous nucleation (chamber studies), implying aqueous uptake is not the only source of isoprene SOA. A clarification is encouraged.**

We have clarified the following sentence so that it now reads: "Isoprene oxidation products can form SOA in the presence and absence of aerosol water (Nguyen et al., 2014; Surratt et al., 2006), though the majority of regional-scale isoprene SOA is currently thought to form through aqueous pathways (Marais et al., 2016)."

**18. Page 2 Line 20: Consider replacing the semicolon by a full stop to break the sentence.**

We have made the change, as suggested.

**19. Page 2 Line 34: I would think this sentence is better fit at the end of the previous paragraph.**

We agree and have moved the sentence.

**20. Page 2 Line 37: Consider substituting 'reaction' with 'oxidation'.**

We have made the suggested change.

**21. Page 2 Line 37: "This includes a major effect on isoprene oxidation chemistry, . . ." what does that mean? Is the major effect simply high and low yield? Or is it differences in chemical pathways? Also consider expanding the literature cited.**

We have changed this sentence to read: "This includes a major effect on the chemical pathway of isoprene oxidation, and on the resulting SOA yield (Ervens et al., 2008; Kroll and Seinfeld, 2008)."

As the Referee notes, we could cite far more studies, not just in this sentence, but in many other places (e.g., see comment #23 below). We already have > 70 references cited, which may be on the high end for an article of this length, but we think appropriate given the large body of literature on isoprene SOA.

**22. Page 3 Line 5: Consider rewording "with our understanding" to "with the understanding".**

We have made the suggested change.

**23. Page 3 Line 9: Consider citing more literature, e.g. (Kroll et al., 2006; Lin et al., 2013; Surratt et al., 2006, 2009).**

We have added (Surratt et al., 2010).

**24. Last paragraph of Introduction: Seems redundant, consider revising or removing.**

We agree with the Referee's suggestion and have removed most of this paragraph. We have edited the first sentence (and moved it to the end of the prior paragraph) so that it now reads: "The aim of this study was to characterize the effects of isoprene and $NO_x$ on aqSOA formed reversibly and irreversibly at a site in the eastern U.S. heavily impacted by biogenic and anthropogenic emissions. "

**25. Page 3 Line 25: Consider using a comma, e.g. ". . .using a mist chamber (MC), and in the particle phase. . .". Furthermore, is a brief description of the MC available? For anyone**

*interested in the technique, which may not be as diffuse as the authors imply, it may be cumbersome to backtrack El-Sayed et al. 2015, then Hennigan et al. 2009, then Cofer and Edahl 1986. Diagrams are encouraged.*

We added the comma, as suggested. In terms of more details on the MC, including a diagram, this MC (or one quite similar) has been described in many prior publications (e.g., (Anderson et al., 2008a; 2008b; Ervens et al., 2011; Hennigan et al., 2008; 2009; Sareen et al., 2016; Spaulding et al., 2002; Zhang et al., 2012) including by our group (El-Sayed et al., 2016; 2015). See our reply to comments #10 and #55, as well.

**26. Page 3 Line 27: Outline the model before explaining what mode it was operated in.**

We have made the suggested change.

**27. Page 3 Line 28: Why is 'dried' in quotes? Given the brief description and lack of diagram, it can be hard for the reader to put words into context.**

This keeps with the convention in our prior publications. We have added the following for clarification: "Note that the $WSOC_{p,dry}$ channel has not been designed to dry particles completely to efflorescence (El-Sayed et al., 2016)."
See also our response to comment #55 below.

**28. Page 3 Line 31: Brand (if any, or if custom made) and dimensions of the parallel plate denuder? What flows can it handle? The gas-phase interferences are not necessarily limited to isoprene oxidation products, is that correct?**

We have added (Sunset Laboratories) to indicate the manufacturer. The Referee is correct that the potential gas-phase interferences would not be limited to isoprene oxidation products; however, we note that such interferences in the PILS have been investigated and are minor, even without the denuder (see Sullivan et al. (2004)).

**29. Page 4 Line 26: The first paragraph of the Results section. . . is it common to take measurements so infrequently? What does the literature recommend?**

Is the Referee suggesting that ~4 weeks of semi-continuous measurements carried out continuously in each season is infrequent? This represents thousands of $WSOC_p$, $WSOC_{p,dry}$, and $WSOC_g$ measurements within each season. The timing is also highly consistent with intensive atmospheric chemistry field campaigns.

**30. Page 4 Line 29: ". . .$WSOC_p$ measurements has been. . ." was it one measurement or multiple? Ensure verb matches the subject of the sentence. If plural, then correct to**

*"...WSOC$_p$ measurements have been...", whereas if singular, correct to "...WSOC$_p$ measurement has been...".*

We changed "has" to "have".

**31.  Page 4 Line 31: Consider removing sentence "In this regard...was formed." as it doesn't add critical information sandwiched between two sentences that by themselves give enough information.**

We agree, and have removed this sentence.

**32.  Page 4 Line 34: Consider having that formula as an equation with a designated equation number. Also, it appears the subscript 'P' is Italicized outside of the bracket, but not inside, and could be corrected. Also, there appears to be a formatting issue with this paragraph in general.**

We have corrected the formatting issues with the paragraph, and the italicized 'p'.  For such a simple formula, whose elements have been clearly defined and discussed prior to this point, it is probably not necessary to designate this as an equation.

**33.  First two paragraphs of Section 3: Consider merging first two paragraphs in one.**

This was a formatting mistake – it has been corrected.

**34.  Page 5 Line 5: Sentence starts with "Figure 1...", yet in Line 8 of the same page, sentence starts with "Fig. 1...". The authors are invited to check for consistency and formatting guidelines of the journal. This may apply for more than one instance.**

"Fig." changed to "Figure".

**35.  Page 5 Line 16: I don't understand the citation to El-Sayed et al., 2016. My understanding is that the values 0.92 and 0.87 for mean WSOCP,dry/WSOCP are from data collected for this manuscript, hence, would not be previously published.**

The mean WSOC$_{p,dry}$/WSOC$_p$ ratio of 0.87 for the summertime data was published in El-Sayed et al. (2016).  Likewise, the WSOC$_{p,dry}$/WSOC$_p$ ratio for the fall was published in El-Sayed et al. (2015).  The previously unpublished data are those from the winter and spring.  Additionally, all of the analyses (e.g., with isoprene) into the factors that affect the WSOC$_{p,dry}$/WSOC$_p$ ratio are new in this manuscript.

**36. Page 5 Line 19: I don't think this sentence belongs here. Aside from this point being stressed before, it is out of place in this paragraph / section. Statements like these should go at the end of the introduction, and they are already included.**

We have changed the sentence so that it now reads: "In the following sections, we characterize the reasons underlying the seasonal differences in $WSOC_{p,dry}/WSOC_p$ shown in Fig. 1."

**37. Page 5 Line 34: The authors could take more care with outlining the Aerosol Mass Spectrometer (AMS) rather than introducing an undefined acronym. In that regard, what is an 'IEPOX factor' and how does it relate to source apportionment techniques/AMS?**

AMS has been defined.

**38. Page 6 Line 29: The authors suggest their diel profile in Fig. S2 is consistent with their data in Figure 3. I would argue that, 3h lag considered, there ought to be an inflection point during the diurnal morning when as WSOCg increases, isoprene decreases. The authors need to address why that inflection in Fig. S2 is not reflected in Fig. 3, arguably indicating the importance of confidence intervals / error bars during the summertime.**

We agree with the Referee's point of adding confidence intervals and/or error bars to several of the Figures (see our response to comment #1 above).  However, with regards to this comment, we disagree with the Referee's suggestion that there should be an inflection point in the $WSOC_g$ (time lagged) diurnal profile.  In the morning, isoprene emissions and OH radical generation both ramp up, and the boundary layer (BL) undergoes rapid expansion.  At some times (07:00 – 10:00 am, local time), isoprene oxidation and BL dilution combine to exceed the effects of fresh isoprene emissions, leading to a decrease in isoprene concentrations.  The inflection point at 10:00 am (local time) comes from a transition where isoprene emissions exceed the loss from oxidation and the effect of dilution, leading to an increase in isoprene concentrations.  There is a fundamental difference in $WSOC_g$; however, in that OH radical oxidation is generally not a loss for most $WSOC_g$ in the same way that it is for isoprene.  $WSOC_g$ is not chemically specific – it is likely made up of hundreds (or more) of different oxygenated organic gases.  Therefore, while oxidation may transform many of the individual $WSOC_g$ compounds, many of these transformations will convert one water-soluble organic gas into another (see (Hodzic et al., 2014)).  Losses of $WSOC_g$ include dry deposition and transformation into SOA, but we do not necessarily expect these losses to exceed $WSOC_g$ production at the same time as isoprene experiences the transition shown in Fig. S2.

**39. Page 6 Line 29: The authors suggest that the chain of reactions leading isoprene to be converted to WSOCg is ~3-5h. While the data is convincing, without air mass trajectories or insolation data, incorporated with statistics, this assertion is slightly weak. Could other VOC or VOC oxidation mechanisms explain WSOC? Is regional terpene, sesquiterpene, or agriculture emission chemistry considered? If it is beyond the scope of the article it should be stated.**

The Referee brings up an excellent point (also made by Referee 3). We have added an explanation for why monoterpene oxidation is not likely contributing to our observations of evaporated $WSOC_p$. The following paragraph is now at the beginning of section 3.3: "During the late spring, the onset of reversible aqSOA formation corresponds to the dramatic increase in isoprene concentrations (Fig. 2). Observations of the Aerodyne aerosol mass spectrometer (AMS) IEPOX factor (Budisulistiorini et al., 2016) and chemical markers for isoprene SOA (Kleindienst et al., 2007) show similarly sharp transitions in the spring and fall in the southeastern U.S. The highest reversible aqSOA levels were observed during the summer when isoprene emissions were at their maximum. Other VOCs, such as monoterpenes, also contribute to SOA in the eastern U.S. (Xu et al., 2015), but monoterpene and isoprene SOA tracers show distinctly different temporal patterns in the eastern U.S. Isoprene SOA peaks during summer, but monoterpene SOA tracers exhibit similar (or lower) concentrations in the summer compared to other seasons (Ding et al., 2008; Kleindienst et al., 2007). Further, monoterpene SOA is typically associated with semi-volatile and less-oxidized OA factors in the AMS analysis (Jimenez et al., 2009; Xu et al., 2015), but $WSOC_p$ is poorly correlated with these factors (Timonen et al., 2013; Xu et al., 2016). On the basis of these prior studies, and the results in Figures 1 and 2, we attribute the reversible aqSOA in Baltimore to isoprene."

**40. Page 7 Lines 13-14: "Consistent with Fig. 3 and Fig. 4. . ." – I do not understand why $WSOC_g$ is strongly correlated with isoprene for lags of 3-5h (Fig. 3) whereas Evaporated $WSOC_p$ is correlated with isoprene for lags of 6-11 h? If evaporated $WSOC_p$ is an example of reversible aqSOA as is $WSOC_g$ by proxy, then if they are produced by the same pathway in the same parcel of air, wouldn't they require the same lag time?**
**If not, and they are two different generation isoprene oxidation products, then why is there a relationship in Fig. S4? This is not clear to me, though perhaps I'm missing something. The following sentence "The above observations suggest that isoprene is strongly linked with the formation of reversible aqSOA in the eastern U.S" therefore does not speak to me.**

Once IEPOX forms from isoprene oxidation, there is still additional time required to form aqSOA. Budisulistiorini et al. (2017) simulate 6- and 12-h processing times for aqSOA to form from IEPOX. We have clarified our discussion about this study so that it now reads: "The (6 to 11) h time lag between isoprene and the evaporated $WSOC_p$ is consistent with the predicted kinetics of IEPOX SOA formation in the eastern U. S. (Budisulistiorini et al., 2017)."

**41. Page 7 Line 20: A simple phrase at the beginning or end of the sentence explaining why the 9h lag was chosen would be helpful. Even though Fig. 4 can by itself be sufficient for an inference, a verbal explanation is helpful.**

We have added the following to the end of this sentence: "since this timing corresponded to the maximum evaporated $WSOC_p$."

*42.  Page 7 Line 22: ". . .it is clear. . ." – as per my comment on Fig. 5, without box-and whiskers, the 'dramatic' decrease is not clear. Upon initial inspection, it would appear most of the data does not exceed 1 ug/m3, thus invalidating the 'dramatic' decrease.*

We have added box-and-whiskers to Fig. 5.  The average difference in evaporated $WSOC_p$ between the lowest and highest bins in Fig. 5 exceeds 1 µg m$^{-3}$.  In Maryland, where the average annual $PM_{2.5}$ concentration is less than 12 µg m$^{-3}$ and the average annual OA concentration is ≈4 µg m$^{-3}$, a difference greater than 1 µg m$^{-3}$ is substantial.  We have changed this sentence to read: "Figure 5 shows that the amount of evaporated $WSOC_p$ decreased substantially with an increase in the $NO_x$/isoprene ratio."

*43.  Page 7 Line 34: Consider rephrasing.*

We have switched the order of the two sentences at the beginning of this paragraph to improve clarity.

*44.  Page 8 Line 15: Awkward phrase: "These results represent, to our knowledge, the first observations to characterize the seasonal occurrence of. . ." consider revising to, e.g., "To the best of our knowledge, observations of seasonal dependence of reversible aqSOA are reported for the first time in this work.".*

We have removed the phrase "to our knowledge".

*45.  Page 8 Line 16: "important implications" has been used 2 out of 3 times in this document at this point. I wonder if it becomes a redundancy. Consider substituting with, e.g., "affect measurement techniques" or something less vague.*

We have changed the sentence to: "The results suggest an important effect on aerosol measurements that implement drying, which may not measure (or may incompletely measure) reversible aqSOA."

*46.  Page 8 Line 21: Consider removing ". . .to confirm this hypothesis."*

We have changed "confirm" to "test."

*47.  Page 8 Line 22: I don't believe the acronym 'AOD' has been defined before by the authors.*

Acronym has been defined.

**48.  Page 9 Line 7: The last sentence is very vague by itself. The paragraph, in general, appears out of place. It is a good point by the authors, but does not seem fit between discussion of aqSOA reversibility on model prediction and discussion of their observations; rather, it can be moved to the end as an anecdotal sentence, or, if elaborated, a paragraph on its own.**

We have replaced this sentence with: "Thus, accounting for the reversible partitioning of water-soluble organic gases to aerosol water would likely improve model predictions of these compounds."

**49.  Page 10 Line 14: If the effect of ALW is more pronounced at low organic concentrations, why is there no discussion about salting out effects, Raoult's law, etc.?**

In this case, we are discussing the effect of ALW on gas-particle partitioning according to Raoult's law (that is what serves as the basis for Pankow's partitioning theory, and the reference cited in this sentence).  Salting in/out would be an effect on aqSOA, which is not the subject of this paragraph.

**50.  Page 10 Line 16: "Our observations show. . ." – if the authors cite their previous publication, I would recommend revising the sentence to "Previous results from our group show. . ." or words to that effect.**

We have made the suggested change.

**51.  Page 10 Line 18: The authors have not defined neither LVOOA nor SVOOA before, unless I missed it.**

Acronyms have been defined.

**52.  Page 10 Line 23: Consider an Oxford comma.**

We have made the suggested change.

**53.  Page 10 Line 25: "They dealt with this problem by. . ." sounds too colloquial. Consider revising.**

We changed "dealt with" to "addressed."

**54.  Page 11 Line 17: Remove first sentence.**

We have made the suggested change.

**Comments on Figures and Tables**

*55. Table S1: Along the same lines of my comments for Page 3 Line 28, this table is not very helpful. It takes a while to understand it. Are the standard deviations for the duration of the study? How often were these measurements made? Would a time series help? Why was the diffusion drier not sized to handle a 90% RH stream and reducing it to <20% RH? What were the dimensions? These details could go in the Supplement (in my opinion).*

We understand the Referee's comment, which is in line with other comments suggesting more experimental details be added to the Methods section and the Supplement (e.g., Comments #10, #27, and #28).  We appreciate the sentiment of having experimental details presented in this paper, reducing the need to refer back to prior papers.  However, we also need to be cognizant of avoiding repetition of descriptions (and in some cases figures) presented in our prior work.  El-Sayed et al. (2016; 2015) present a detailed discussion of the methods we use in this work (both present instrument schematics), including discussions relevant to this Referee comment.  For example, the 2$^{nd}$ paragraph of the "Materials and Methods" section of El-Sayed et al. (2016) includes:

"*The goal for the $WSOC_{p,dry}$ measurement was not to remove all particle bound water, but rather to approximate the lowest RH that particles may be exposed to in ambient air during the study period to simulate "natural" drying processes (Supporting Information Figure S2). The dried channel included a silica gel diffusion dryer, which was made in-house similar to commercial models (e.g., TSI model 3062). $WSOC_p$ losses through the 3- way valve and through the dried channel were evaluated prior to the start of the sampling period and were found to be negligible (Supporting Information Figure S3). The dryer was replaced daily and its efficiency was checked with an orange silica gel color-indicator as well as an RH sensor (Omega, RHUSB) that measured the RH of air exiting the dryer.*"

*56. Figure 1: With the understanding that the authors composed a box-and-whiskers diagram to visualize their data, can something be done about the x-axis potentially misleading a reader that all five data are not evenly spaced across the year? If not, that is OK in my view, but if the data can be displayed with the x-axis being more akin to DateTime, it would better visualize (in my opinion) the seasonal cycles the authors wish to present.*

Table 1 presents the dates that correspond to the seasonal labels in Figure 1.  In the final version of the published paper, we will request that Table 1 and Figure 1 appear on the same page so that readers can easily locate these dates.

*57. Figure 2: Upon reading the caption, this is an annual profile averaged across 5 years. I would request the data be replotted using markers and lines, at least, and ideally with some form of confidence intervals to reflect the averaged data. While the point of the authors is that*

*isoprene is high during the summer months, the data can be presented with a little more rigor and care. If data from MDE comes like this, the authors can state it.*

We agree with the Referee's suggestion, and have updated Figure 2 accordingly.

**58. Figure 3: If the authors claim that their calculation (or rather, literature review) of isoprene lifetime to OH oxidation is on the order of 1-2h, then this figure really requires at least vertical error bars. While the median WSOCg does correlate with isoprene mixing ratios at lag times between 3-5 h, other types of statistics are encouraged for the argument to be valid.**

We have added a supplemental figure to support Figure 3 that shows the individual data points and box and whiskers for one of the lag times. We also point the Referee to our extensive discussion this figure in Section 3.3, especially: "Overall, this suggests that fresh isoprene emissions take about (3 to 5) h to form $WSOC_g$ in an urban environment during typical summertime conditions. Note that the measurement of $WSOC_g$ only includes compounds with effective Henry's law constants above $\sim 10^3$ M/atm (Spaulding et al., 2002), so the MC does not efficiently sample many first-generation isoprene oxidation products, such as methacrolein ($K_H = 4 \times 10^0$ M/atm) or methyl vinyl ketone ($K_H = 4 \times 10^1$ M/atm) (Sander, 2015). "

**59. Figure 5: Consider visuals, at least on the x-axis, to show regime of polluted vs clean air (low values on the x-axis are clean; high values are polluted). Also, if formatting permits, vertical box plots could help visualize the binning. In my opinion, the graph is very misleading otherwise.**

We have added box plots to Figure 5.

**Cited References**

Anderson, C., Dibb, J.E., Griffin, R.J., Bergin, M.H., Simultaneous measurements of particulate and gas-phase water-soluble organic carbon concentrations at remote and urban-influenced locations, *Geophysical Research Letters* **35**(2008a).

Anderson, C.H., Dibb, J.E., Griffin, R.J., Hagler, G.S.W., Bergin, M.H., Atmospheric water-soluble organic carbon measurements at Summit, Greenland, *Atmospheric Environment* **42**(2008b), pp. 5612-5621.

Anderson, D.C. *et al.*, Measured and modeled CO and NO y in DISCOVER-AQ: An evaluation of emissions and chemistry over the eastern US, *Atmospheric environment* **96**(2014), pp. 78-87.

Beyersdorf, A.J. *et al.*, The impacts of aerosol loading, composition, and water uptake on aerosol extinction variability in the Baltimore–Washington, D.C. region, *Atmos. Chem. Phys.* **16**(2016), pp. 1003-1015.

Budisulistiorini, S.H. *et al.*, Seasonal characterization of submicron aerosol chemical composition and organic aerosol sources in the southeastern United States: Atlanta, Georgia,and Look Rock, Tennessee, *Atmos. Chem. Phys.* **16**(2016), pp. 5171-5189.

Budisulistiorini, S.H. *et al.*, Simulating Aqueous-Phase Isoprene-Epoxydiol (IEPOX) Secondary Organic Aerosol Production During the 2013 Southern Oxidant and Aerosol Study (SOAS), *Environmental Science & Technology* **51**(2017), pp. 5026-5034.

Ding, X. *et al.*, Spatial and seasonal trends in biogenic secondary organic aerosol tracers and water-soluble organic carbon in the southeastern United States, *Environmental science & technology* **42**(2008), pp. 5171-5176.

El-Sayed, M.M.H., Amenumey, D., Hennigan, C.J., Drying-Induced Evaporation of Secondary Organic Aerosol during Summer, *Environmental Science & Technology* **50**(2016), pp. 3626-3633.

El-Sayed, M.M.H., Wang, Y.Q., Hennigan, C.J., Direct atmospheric evidence for the irreversible formation of aqueous secondary organic aerosol, *Geophysical Research Letters* **42**(2015), pp. 5577-5586.

Ervens, B. *et al.*, Secondary organic aerosol yields from cloud-processing of isoprene oxidation products, *Geophysical Research Letters* **35**(2008).

Ervens, B., Turpin, B.J., Weber, R.J., Secondary organic aerosol formation in cloud droplets and aqueous particles (aqSOA): a review of laboratory, field and model studies, *Atmos. Chem. Phys.* **11**(2011), pp. 11069-11102.

Fu, T.M. *et al.*, Global budgets of atmospheric glyoxal and methylglyoxal, and implications for formation of secondary organic aerosols, *Journal of Geophysical Research-Atmospheres* **113**(2008).

Guenther, A.B. *et al.*, The Model of Emissions of Gases and Aerosols from Nature version 2.1 (MEGAN2.1): an extended and updated framework for modeling biogenic emissions, *Geosci. Model Dev.* **5**(2012), pp. 1471-1492.

Hennigan, C.J., Bergin, M.H., Dibb, J.E., Weber, R.J., Enhanced secondary organic aerosol formation due to water uptake by fine particles, *Geophysical Research Letters* **35**(2008).

Hennigan, C.J., Bergin, M.H., Russell, A.G., Nenes, A., Weber, R.J., Gas/particle partitioning of water-soluble organic aerosol in Atlanta, *Atmospheric Chemistry and Physics* **9**(2009), pp. 3613-3628.

Henze, D.K., Seinfeld, J.H., Global secondary organic aerosol from isoprene oxidation, *Geophysical Research Letters* **33**(2006).

Hodzic, A. *et al.*, Volatility dependence of Henry's law constants of condensable organics: Application to estimate depositional loss of secondary organic aerosols, *Geophysical Research Letters* **41**(2014), pp. 4795-4804.

Jimenez, J.L. *et al.*, Evolution of Organic Aerosols in the Atmosphere, *Science* **326**(2009), p. 1525.

Kim, P.S. *et al.*, Sources, seasonality, and trends of southeast US aerosol: an integrated analysis of surface, aircraft, and satellite observations with the GEOS-Chem chemical transport model, *Atmos. Chem. Phys.* **15**(2015), pp. 10411-10433.

Kleindienst, T.E. *et al.*, Estimates of the contributions of biogenic and anthropogenic hydrocarbons to secondary organic aerosol at a southeastern US location, *Atmospheric Environment* **41**(2007), pp. 8288-8300.

Kroll, J.H., Seinfeld, J.H., Chemistry of secondary organic aerosol: Formation and evolution of low-volatility organics in the atmosphere, *Atmospheric Environment* **42**(2008), pp. 3593-3624.

Liu, J. *et al.*, Efficient isoprene secondary organic aerosol formation from a non-IEPOX pathway, *Environmental science & technology* **50**(2016), pp. 9872-9880.

Marais, E.A. *et al.*, Aqueous-phase mechanism for secondary organic aerosol formation from isoprene: application to the southeast United States and co-benefit of SO2 emission controls, *Atmos. Chem. Phys.* **16**(2016), pp. 1603-1618.

Nguyen, T.B. *et al.*, Organic aerosol formation from the reactive uptake of isoprene epoxydiols (IEPOX) onto non-acidified inorganic seeds, *Atmos. Chem. Phys.* **14**(2014), pp. 3497-3510.

Nguyen, T.B. *et al.*, Rapid deposition of oxidized biogenic compounds to a temperate forest, *Proceedings of the National Academy of Sciences* **112**(2015), pp. E392-E401.

Pfister, G.G. *et al.*, Contribution of isoprene to chemical budgets: A model tracer study with the NCAR CTM MOZART-4, *Journal of Geophysical Research: Atmospheres* **113**(2008).

Psichoudaki, M., Pandis, S.N., Atmospheric Aerosol Water-Soluble Organic Carbon Measurement: A Theoretical Analysis, *Environmental Science & Technology* **47**(2013), pp. 9791-9798.

Pye, H.O.T. *et al.*, On the implications of aerosol liquid water and phase separation for organic aerosol mass, *Atmos. Chem. Phys.* **17**(2017), pp. 343-369.

Pye, H.O.T. *et al.*, Epoxide Pathways Improve Model Predictions of Isoprene Markers and Reveal Key Role of Acidity in Aerosol Formation, *Environmental Science & Technology* **47**(2013), pp. 11056-11064.

Sander, R., Compilation of Henry's law constants (version 4.0) for water as solvent, *Atmos. Chem. Phys.* **15**(2015), pp. 4399-4981.

Sareen, N. *et al.*, Identifying precursors and aqueous organic aerosol formation pathways during the SOAS campaign, *Atmos. Chem. Phys.* **16**(2016), pp. 14409-14420.

Sareen, N., Waxman, E.M., Turpin, B.J., Volkamer, R., Carlton, A.G., Potential of Aerosol Liquid Water to Facilitate Organic Aerosol Formation: Assessing Knowledge Gaps about Precursors and Partitioning, *Environmental Science & Technology* **51**(2017), pp. 3327-3335.

Spaulding, R.S., Talbot, R.W., Charles, M.J., Optimization of a Mist Chamber (Cofer Scrubber) for Sampling Water-Soluble Organics in Air, *Environmental Science & Technology* **36**(2002), pp. 1798-1808.

Sullivan, A.P. *et al.*, A method for on-line measurement of water-soluble organic carbon in ambient aerosol particles: Results from an urban site, *Geophysical Research Letters* **31**(2004).

Surratt, J.D. *et al.*, Reactive intermediates revealed in secondary organic aerosol formation from isoprene, *Proceedings of the National Academy of Sciences* **107**(2010), pp. 6640-6645.

Surratt, J.D. *et al.*, Chemical Composition of Secondary Organic Aerosol Formed from the Photooxidation of Isoprene, *The Journal of Physical Chemistry A* **110**(2006), pp. 9665-9690.

Timonen, H. *et al.*, Characteristics, sources and water-solubility of ambient submicron organic aerosol in springtime in Helsinki, Finland, *Journal of Aerosol Science* **56**(2013), pp. 61-77.

Travis, K.R. *et al.*, Why do models overestimate surface ozone in the Southeast United States?, *Atmos. Chem. Phys.* **16**(2016), pp. 13561-13577.

Weber, R.J. *et al.*, A study of secondary organic aerosol formation in the anthropogenic-influenced southeastern United States, *Journal of Geophysical Research-Atmospheres* **112**(2007).

Wong, J.P.S., Zhou, S., Abbatt, J.P.D., Changes in Secondary Organic Aerosol Composition and Mass due to Photolysis: Relative Humidity Dependence, *The Journal of Physical Chemistry A* **119**(2015), pp. 4309-4316.

Xu, L. *et al.*, Effects of anthropogenic emissions on aerosol formation from isoprene and monoterpenes in the southeastern United States, *Proceedings of the National Academy of Sciences* **112**(2015), pp. 37-42.

Xu, L., Guo, H., Weber, R.J., Ng, N.L., Chemical characterization of water-soluble organic aerosol in contrasting rural and urban environments in the southeastern United States, *Environmental science & technology* **51**(2016), pp. 78-88.

Ying, Q., Li, J., Kota, S.H., Significant Contributions of Isoprene to Summertime Secondary Organic Aerosol in Eastern United States, *Environmental Science & Technology* **49**(2015), pp. 7834-7842.

Yu, K. *et al.*, Sensitivity to grid resolution in the ability of a chemical transport model to simulate observed oxidant chemistry under high-isoprene conditions, *Atmos. Chem. Phys.* **16**(2016), pp. 4369-4378.

Zhang, X.L. *et al.*, On the gas-particle partitioning of soluble organic aerosol in two urban atmospheres with contrasting emissions: 1. Bulk water-soluble organic carbon, *Journal of Geophysical Research-Atmospheres* **117**(2012).

---

## Author Comment (AC2) · 1 Dec 2017

**Response to Reviews**

We thank the reviewers for their detailed comments and helpful suggestions. We have addressed each comment below, with the Referee comment in bold italicized text, our response in plain text, and any manuscript changes noted in red text. In addition, the revised manuscript with changes marked up has been attached to the end of our response to Referee 3.

**Anonymous Referee #2**

**General comments:**

This work examines aqueous SOA, both reversible (able to evaporate upon drying) and irreversible, in the Eastern US using measurements of water-soluble compounds in both the gas and particle phase. Additional measurements (isoprene, NOx) are used to infer that the reversible SOA is a result of isoprene epoxydiol (IEPOX) uptake to an aqueous medium. This paper examines an important issue with implications for what controls IEPOX SOA formation. However, to further their conclusions it would be good to demonstrate stronger connections between isoprene and the reversible SOA since no chemical identity beyond WSOCp (particulate WSOC) and WSOCp, dry (dried WSOC) is known for the organic aerosol. The major pieces of evidence for IEPOX being the precursor to reversible SOA come from NOx and isoprene concentrations and time lag analysis. The WSOCp peaks 9 hours after isoprene (consistent with IEPOX being 2nd generation plus an additional lag), and the reversible SOA is highest when NOx/Isoprene is lowest which is consistent with our understanding of IEPOX formation in the gas-phase. However, formation of IEPOX may not be the limiting factor for IEPOX SOA formation (sulfate and its influence on particle surface area/volume as well as acidity may be responsible). In addition, other aspects of the ambient atmosphere are changing in addition to NOx and isoprene as a function of season. Two areas that could be furthered include:

**1. Can mass closure be reached in terms of how much isoprene is present and the amount of WSOCp and WSOCg? E.g. Page 6 line 25: Do you get mass closure if you assume 5 ppbC of isoprene reacted forms 2 ugC/m3 IEPOX?**

We agree with the reviewer that such an analysis would be quite interesting; however, we are not able to quantitatively link the reacted isoprene and formed  $WSOC_p$ . This is an inherent limitation of the  $WSOC_p$  and isoprene measurements at different locations. We have added detailed discussion about this point in the Methods section (see also our response to Referee 3, comment #1). Based on our methods and analyses, we are only able show a strong link between isoprene and reversible aqSOA: attempting a mass closure analysis would be too speculative and would have a prohibitively high uncertainty.

2. Is the reversible IEPOX SOA just dissolved IEPOX or is it a reversibly formed reaction product? Are the levels of reversible IEPOX SOA consistent with dissolved IEPOX? Sareen et al. 2017 indicate dissolved IEPOX alone is a very small concentration (especially compared to IEPOX SOA from AMS PMF analysis).

The reviewer brings up an excellent point. We have clarified several points in the text: "Note that Sareen et al. (2017) predict very low dissolved IEPOX in the eastern U.S. during summer ( $< 0.01 \ \mu g \ m^{-3}$ ), suggesting reversibly formed reaction products are the dominant contributors to reversible aqSOA." and also: "For example, it is unclear how the instruments employed by Lopez-Hilfiker et al. (2016) and Hu et al. (2016) respond to reversible IEPOX reaction products present in the aqueous phase."

**3. Were other proxies for chemistry besides NOx/Isoprene examined? Page 7, near line 10: Is the diurnal variation in sulfate involved in IEPOX SOA?**

In the eastern U.S., sulfate typically shows little variability throughout the day (e.g., Fig. S2b in (Xu et al., 2015)). However, sulfate does play a critical role in IEPOX chemistry, so we have added substantial analysis and discussion related to this point. See our response to comments #4 and #5 below.

**4. Figure 7 shows seasonality in the WSOCp,dry/WSOCp ratio consistent with changes in NOx/Isoprene. What else changes as a function of season that could also explain the ratio? Oxidants? How is ALW changing? If the horizontal axis was sulfate or SOx divided by isoprene would it show the same behavior?**

ALW does not vary significantly across our late spring, summer, and fall sampling periods ( $\approx$ 20% differences, Paper in preparation). Ozone does exhibit a strong seasonal pattern in the eastern U.S., increasing in the spring, peaking during summer, decreasing during fall, with a minimum in winter. In terms of SOA contributions, ozone reactions with monoterpenes are far more important than ozone reactions with isoprene (Xu et al., 2015). However, monoterpene SOA is produced year round, and does not peak during summer in the eastern U.S. (see also our response to Referee 3, comment #2 for more detailed discussion of this point). Sulfate strongly affects isoprene SOA; however, we do not have sufficient sulfate data to incorporate such an analysis in the present study. SO2 and sulfate are not correlated in Baltimore (R2 = 0.06), indicating that a figure analogous to Fig. 7 but instead with SO2/isoprene on the x-axis would not provide the desired insight into the effects of sulfate. Studies show that sulfate and NOx both affect isoprene SOA (e.g., (de Sá et al., 2017)), so even if there are other important species that we do not consider here, our analysis of NOx/isoprene is still valid.

**Other Specific Comments:**

5. Page 1: Lines 23-24 indicate that the trend towards lower NOx/Isoprene ratios may mean more IEPOX SOA in the future. Given the dependence of IEPOX SOA on sulfate, wouldn't we expect this pathway to decrease with decreasing sulfate levels in the future as demonstrated by Marais et al. 2017 ERL

(http://iopscience.iop.org/article/10.1088/1748-9326/aa69c8/meta)?

The Referee brings up an excellent point. We have removed the sentence from the abstract, and have added the following paragraph to the Conclusions: "Predictions of future  $NO_x$  and isoprene emissions in response to regulations, technology, and climate change also suggest that this process may increase in importance going forward. Such an inference is complicated by concurrent reductions in SO2 emissions in the U.S. and other developed nations. The consequent decreases in sulfate may offset the effects of  $NO_x$  reductions on isoprene SOA (de Sá et al., 2017). However, we stress that prior studies into the  $NO_x$ -sulfate-isoprene system have not systematically determined how these species affect the reversibility of isoprene SOA. Therefore, while we hypothesize that future decreases in  $NO_x$  and increases in isoprene will increase reversible isoprene SOA (or at least the reversible fraction), the role of changing sulfate will also need to be considered. Future laboratory and modeling studies will be needed to address this question directly."

**6. Page 1: Line 29 indicates isoprene is the dominant SOA precursor in summer. I would define dominant as responsible for >= 50% of SOA. Hu et al. 2015 ACP (https://doi.org/10.5194/acp-15-11807-2015) indicate isoprene (or IEPOX) is responsible for 17% to 36% of Southeast US SOA. So while it is important, it is not dominant.**

The Referee is correct that measurements and models constrain the IEPOX contribution to SOA in the eastern U.S. during summer to less than 50%. However, isoprene also forms SOA that is not necessarily associated with the IEPOX factor identified by the AMS. The studies we have cited here both predict that isoprene is the dominant SOA precursor, contributing more than 50% of SOA in the southeastern U.S during summer. These predictions have uncertainties, though, and to acknowledge this, we have changed the text to read: "In the southeastern United States, isoprene is likely the dominant SOA precursor during summer (Kim et al., 2015; Ying et al., 2015)."

7. Page 2: Lines 21-23: I would characterize both Marais et al. 2016 and Pye et al. 2013 as irreversible IEPOX uptake since both use a reactive uptake formulation. The major difference between Marais et al. and Pye et al. is the Henry's law coefficient which leads to different amounts of IEPOX SOA. They also simulated different years. Budisulistiorini et al. 2017 has shown that reversible (simpleGAMMA, McNeill et al. 2012) and irreversible (CMAQ, Pye et al. 2013) models of IEPOX uptake can agree when the parameters going into them are identical (for ~6 hours of processing time).

We agree with the reviewer's comment, and have removed this sentence.

**8. Page 3: Line 17-18: which instruments may not measure reversible SOA?**

These instruments may include: AMS, f(RH),  $PM_{2.5}$  mass concentrations and others. We have a full discussion about instruments and methods that might not be able to measure reversible aqSOA due to particle drying in our previous publication (El-Sayed et al., 2016).

**9. Page 4: Near line 30: Can you clarify the relationship between WSOCp and aqSOA? What fraction of WSOCp is aqSOA? How was aqSOA identified?**

We have added the following to the Methods section: "WSOCp is operationally defined based upon the solubilites of the organics, themselves, and the level of dilution employed for the analysis (Psichoudaki and Pandis, 2013). In the eastern U.S., the WSOCp measurement is a surrogate for SOA, especially during summer (Weber et al., 2007). The measurement includes SOA formed through absorptive partitioning and through aqueous-mediated pathways (aqSOA). We consider any WSOCp that evaporates with drying to be reversible aqSOA, since this material exists in the condensed phase because of the aerosol water and partitions back to the gas phase when the water evaporates."

aqSOA is identified based upon the relationship between  $F_p$  ( $F_p = WSOC_p/(WSOC_p + WSOC_g)$ ) and relative humidity, according to Hennigan et al. (2008) and El-Sayed et al. (2016, 2015). We have another paper in preparation that focuses on the other question (*What fraction of WSOCp is aqSOA?*).

**10. Page 8: Line 35: How much higher is the fraction of reversible aqSOA? Insert value.**

We have changed the text so that it now reads: "The results in Fig. 1 show that  $\approx 10$  to 15 % of the total WSOCp evaporates with drying during the late spring and summer, on average. This suggests that the fraction of aqSOA that is formed reversibly is much higher than 15% since the measurement of WSOCp includes compounds formed through uptake to aqueous particles (aqSOA) and compounds formed through traditional SOA pathways."

**11. Page 8: Line 36: For the range of 0-60%, what is the typical value (mean, median, or similar)?**

We have changed the text so that it now reads: "The results in Fig. 1 show that  $\approx 10$  to 15 % of the total WSOCp evaporates with drying during the late spring and summer, on average. This suggests that the fraction of aqSOA that is formed reversibly is much higher than 15% since the measurement of WSOCp includes compounds formed through uptake to aqueous particles (aqSOA) and compounds formed through traditional SOA pathways."

12. Page 9: Line 18-22: I am unclear as to whether or not the work of Wong et al., 2015 is atmospherically relevant if their experiments did not produce SOA from IEPOX. D'Ambro et al. 2017 ES&T (http://pubs.acs.org/doi/abs/10.1021/acs.est.7b00460) demonstrates that IEPOX is the atmospherically relevant pathway to isoprene SOA and laboratory experiments with unrealistic concentrations may be activating pathways that are not important in the atmosphere.

We agree with the reviewer's comment and have added the following to our discussion: "Although the experiments of Wong et al. (2015) were performed in a chemical regime where IEPOX formation is favored, it did not contribute to the SOA in their experiments due to high OH levels. Given the absence of IEPOX-SOA in the experiments of Wong et al. (2015), the atmospheric relevance of their results may be questionable. However, the uptake of other, non-IEPOX, low-NOx oxidation products may explain such observations (Liu et al., 2016; Riva et al., 2016)."

**Cited References**

- El-Sayed, M.M.H., Amenumey, D., Hennigan, C.J., Drying-Induced Evaporation of Secondary Organic Aerosol during Summer, *Environmental Science & Technology* **50**(2016), pp. 3626-3633.
- El-Sayed, M.M.H., Wang, Y.Q., Hennigan, C.J., Direct atmospheric evidence for the irreversible formation of aqueous secondary organic aerosol, *Geophysical Research Letters* **42**(2015), pp. 5577-5586.
- Hennigan, C.J., Bergin, M.H., Dibb, J.E., Weber, R.J., Enhanced secondary organic aerosol formation due to water uptake by fine particles, *Geophysical Research Letters* **35**(2008).
- Hu, W. *et al.*, Volatility and lifetime against OH heterogeneous reaction of ambient isoprene-epoxydiols-derived secondary organic aerosol (IEPOX-SOA), *Atmos. Chem. Phys.* **16**(2016), pp. 11563-11580.
- Kim, P.S. *et al.*, Sources, seasonality, and trends of southeast US aerosol: an integrated analysis of surface, aircraft, and satellite observations with the GEOS-Chem chemical transport model, *Atmos. Chem. Phys.* **15**(2015), pp. 10411-10433.
- Liu, J. et al., Efficient isoprene secondary organic aerosol formation from a non-IEPOX pathway, *Environmental science & technology* **50**(2016), pp. 9872-9880.
- Lopez-Hilfiker, F.D. *et al.*, Molecular Composition and Volatility of Organic Aerosol in the Southeastern U.S.: Implications for IEPOX Derived SOA, *Environmental Science & Technology* **50**(2016), pp. 2200-2209.
- Psichoudaki, M., Pandis, S.N., Atmospheric Aerosol Water-Soluble Organic Carbon Measurement: A Theoretical Analysis, *Environmental Science & Technology* **47**(2013), pp. 9791-9798.
- Riva, M. *et al.*, Chemical characterization of secondary organic aerosol from oxidation of isoprene hydroxyhydroperoxides, *Environmental science & technology* **50**(2016), pp. 9889-9899.
- Sareen, N., Waxman, E.M., Turpin, B.J., Volkamer, R., Carlton, A.G., Potential of Aerosol Liquid Water to Facilitate Organic Aerosol Formation: Assessing Knowledge Gaps about Precursors and Partitioning, *Environmental Science & Technology* 51(2017), pp. 3327-3335.
- de Sá, S.S. *et al.*, Influence of urban pollution on the production of organic particulate matter from isoprene epoxydiols in central Amazonia, *Atmospheric Chemistry and Physics* **17**(2017), pp. 6611-6629.
- Weber, R.J. *et al.*, A study of secondary organic aerosol formation in the anthropogenic-influenced southeastern United States, *Journal of Geophysical Research-Atmospheres* **112**(2007).
- Wong, J.P.S., Zhou, S., Abbatt, J.P.D., Changes in Secondary Organic Aerosol Composition and Mass due to Photolysis: Relative Humidity Dependence, *The Journal of Physical Chemistry A* 119(2015), pp. 4309-4316.
- Xu, L. *et al.*, Effects of anthropogenic emissions on aerosol formation from isoprene and monoterpenes in the southeastern United States, *Proceedings of the National Academy of Sciences* **112**(2015), pp. 37-42.
- Ying, Q., Li, J., Kota, S.H., Significant Contributions of Isoprene to Summertime Secondary Organic Aerosol in Eastern United States, *Environmental Science & Technology* **49**(2015), pp. 7834-7842.

---

## Author Comment (AC3) · 1 Dec 2017

**Response to Reviews**

We thank the reviewers for their detailed comments and helpful suggestions. We have addressed each comment below, with the Referee comment in bold italicized text, our response in plain text, and any manuscript changes noted in red text. In addition, the revised manuscript with changes marked up has been attached to the end of our response to Referee 3.

**Anonymous Referee #3**

*General comments:*
*This article examines the influence of NOx on the reversibility of aqueous SOA. The paper analyzed the irreversible and reversible water-soluble organic carbon (WSOC) in the particle phase, as well as WSOC in the gas phase from a site in the Eastern US. By correlating the gas phase and particle phase WSOC with isoprene concentration measured at a nearby site (~20 km away), the author believes that IEPOX uptake is responsible for the reversible aqueous SOA. By correlating the NOx/isoprene ratio vs. the percentage of reversible aqueous SOA of total particle phase WSOC, the author suggests that low NOx/isoprene ratios seem to enhance reversible aqueous SOA formation, which agrees with the author's assumption that IEPOX is the main reason for reversible aqSOA formed in late spring and summer.*
*This study provides a nice perspective of how reversible WSOC could affect the SOA budget and how drying the aerosol before performing field measurement could neglect reversible WSOC. The work performed to attribute the sources of such reversible WSOC by performing a time lag correlation on the isoprene concentration is interesting, but the lack of further analysis on the molecular composition of the gas and particle phase WSOC weakens the conclusion.*

*Overall, this article provides a unique perspective on the importance of WSOC in SOA with enough scientific content and novelty to be published in ACP. However, the conclusion that IEPOX is the likely cause of reversible WSOC in aqSOA is not strong enough and the logic between sentences in some paragraphs is not clear. The author needs to address the following issues and refine the wording before being published in ACP.:*

*1. The sampling site of WSOC (Baltimore) is ~20 km away from the site sampling isoprene and NOx (Essex site), so whether the Essex site can be representative of the Baltimore site is a questionable part of this study, especially when the Baltimore site is heavily influenced by anthropogenic emissions and the author showed up to 11 hour lag comparison between the two sites. Back trajectory data would be better to use in this paper to justify the result, in order to (1) either prove that Baltimore is downwind of the Essex Site, (2) or to filter out those data when Baltimore is not downwind of the Essex site.*

We agree with the Referee that more discussion is needed to justify our use of data from different sites (Referee #1 had a similar comment). We have added the following discussion to Section 2 to support our methods: "A key assumption employed in this analysis is that the WSOC measurements made at UMBC are representative of conditions at Essex, the location of

the NO$_x$ and isoprene measurements.  Aerosol concentrations in the Baltimore-Washington region are spatially uniform over tens of kilometers (Beyersdorf et al., 2016).  Further, WSOC$_p$ concentrations exhibit small spatial variations across urban-to-rural gradients during the summertime (Weber et al., 2007).  These prior analyses showed that aerosol concentrations, and in particular WSOC, were not dependent on wind direction.  Isoprene emissions in the eastern U.S. are regional in nature, due to the expansive coverage of broadleaf forests (Guenther et al., 2012; Pye et al., 2013).  NO$_x$ emissions are spatially segregated from those of isoprene, and are far more localized.  However, the isoprene-NO$_x$ chemical regime (high- or low-NO$_x$) in the eastern U.S. is generally well-represented with model resolution of 28 x 28 km, suggesting that the chemistry occurring on small scales, such as in individual power plant plumes, does not significantly affect the regional isoprene-NO$_x$ regime (Yu et al., 2016).  NO$_x$ concentrations at Essex (20 km ENE of UMBC) and HU-Beltsville (35 km SSW of UMBC) are strongly correlated (R = 0.89, Fig. S6), likely due to the overwhelming contribution of mobile source emissions along the heavily-traveled I-95 corridor to the region (Anderson et al., 2014).  Together, this supports our analysis into the effects of isoprene and NO$_x$ on reversible aqSOA using the measurements described above.”

*2. It is difficult to make a strong argument that IEPOX is the main reason for the reversible WSOC in the particle phase without chemical characterization. Other BVOCs (such as monoterpenes) can also form water-soluble components that were shown to enhance SOA mass at high RH (Prisle et al., 2010), and their reaction mechanisms are also sensitive to NO$_x$ concentration (Wildt et al.). Even though IEPOX may seem to be a more likely compound for reversible aqSOA for this study, the author needs to provide stronger evidence to rule out other possibilities, such as a correlation plot between particle WSOC vs. time lagged monoterpene concentration.*

The Referee brings up an excellent point (also made by Referee 1).  We have added an explanation for why monoterpene oxidation is not likely contributing to our observations of evaporated WSOC$_p$.  The following paragraph is now at the beginning of section 3.3: “During the late spring, the onset of reversible aqSOA formation corresponds to the dramatic increase in isoprene concentrations (Fig. 2).  Observations of the Aerodyne aerosol mass spectrometer (AMS) IEPOX factor  (Budisulistiorini et al., 2016) and chemical markers for isoprene SOA (Kleindienst et al., 2007) show similarly sharp transitions in the spring and fall in the southeastern U.S.  The highest reversible aqSOA levels were observed during the summer when isoprene emissions were at their maximum.  Other VOCs, such as monoterpenes, also contribute to SOA in the eastern U.S. (Xu et al., 2015), but monoterpene and isoprene SOA tracers show distinctly different temporal patterns in the eastern U.S.  Isoprene SOA peaks during summer, but monoterpene SOA tracers exhibit similar (or lower) concentrations in the summer compared to other seasons (Ding et al., 2008; Kleindienst et al., 2007).  Further, monoterpene SOA is typically associated with semi-volatile and less-oxidized OA factors in the AMS analysis (Jimenez et al., 2009; Xu et al., 2015), but WSOC$_p$ is poorly correlated with these factors (Timonen et al., 2013; Xu et al., 2016). On the basis of these prior studies, and the results in Figures 1 and 2, we attribute the reversible aqSOA in Baltimore to isoprene.”

*3. Page 3, line 7, the author listed the ACP paper by Chan Miller et al. (2017) to show that glyoxal is formed in both low- and high- NOx pathways while IEPOX is mainly formed in the low NOx pathway. Therefore the author states that the correlation between NOx/isoprene can be attributed to IEPOX. But, the yield of glyoxal in high and low NOx conditions are different. Chan Miller et al. shows in his paper (Figure 2) that glyoxal formed from isoprene oxidation has a higher yield at low NOx condition compared with a high NOx condition. Therefore, I believe the increase of reversible aqSOA could be at least partially attributed to glyoxal. The author needs to specify all these possibilities in the paper rather than attributing the reversible aqSOA solely on IEPOX. The conclusion in the abstract as well as throughout the paper is too strong and needs to be revised.*

The modeling study of Chan Miller et al. (2017) predicts that, in the eastern U.S. during summer, glyoxal production from isoprene is almost equal between high- and low-$NO_x$ pathways (see their Figure 1 with quantitative contributions from each pathway corresponding to the eastern U.S. in June and July). We have clarified the text so that it now reads: "During the summer, model predictions suggest that glyoxal production from isoprene occurs almost equally through low- and high-$NO_x$ pathways in the eastern U.S. (Chan Miller et al., 2017)." However, we agree with the Referee's comment that stronger justification is needed to support the explanation that IEPOX is largely responsible for our observed reversible aqSOA. We have added substantial discussion related to this point (see our response to comments #2, #13, and #14).

*4. Besides comparing WSOC$_{p, dry}$/WSOC$_p$ with isoprene concentration, has the author studied the influence of ambient humidity on WSOC$_{p, dry}$/WSOC$_p$?*

Yes, we have looked at the influence of ambient humidity on $WSOC_{p, dry}$/$WSCO_p$ in the fall (El-Sayed et al., 2015) and during the summer (El-Sayed et al., 2016) and the full seasonal characterization is detailed in a paper in preparation.

*5. Moreover, from TableS1, it seems that when ambient RH=80%, the RHs of the samples passed through the silica gel dryer were consistently higher in the summer time than in the winter time. What are the reasons and would that cause artifacts of the results?*

This is probably due to differences in the absolute humidity levels during both seasons. With the $WSOC_p$/$WSOC_{p,dry}$ system, our goal is not to dry the particles completely. Rather, it is to mimic the drying that particles typically undergo near the surface as a result of ambient meteorological variations. The dryer does a reasonable job of this: during the summer, the RH through the dryer suggests that the particles lose most ALW but do not dry completely (i.e., at RH of 35 – 40% there is still ALW). During the winter, however, the RH through the dryer suggests that the particles are dried completely. This is supported by an ambient study at a similar location, where it was observed that particles during the summer almost always contained some water (even at the lowest ambient RH levels), while particles during the winter were often dry (Khlystov et al., 2005). This has been added to the text in the Supplement: "Note, differences in the RH through the dryer are likely due to differences in absolute humidity levels within each season."

*6. Because IEPOX usually undergoes reactive uptake with high acidity aerosols (Gaston et al., 2014; Riedel et al., 2015), people have been assuming that isoprene-derived SOA is not very important in low acidity aerosols. However, the result presented in this study shows the importance of isoprene-derived SOA even for low acidity aerosol particles, especially when there is an amount of liquid water in the aerosol so IEPOX can have reversible partitioning. The author should probably talk about the importance of this aspect in the atmospheric implication section as well.*

Based on several recent studies (Budisulistiorini et al., 2016; Guo et al., 2017; 2016; 2015; Weber et al., 2016; Xu et al., 2015), we do not believe that our results offer a different perspective on the role of particle acidity in forming isoprene SOA. These studies suggest that (1) particles in the eastern U.S. are quite acidic (pH < 2) most of the time, and (2) the pH is low enough that acidity is not likely a limiting factor for IEPOX SOA formation in the eastern U.S. What is quite interesting is that it appears isoprene forms reversible SOA, even in the presence of acidic particles. We have added the following discussion about this point: "In addition to $NO_x$, sulfate also strongly affects SOA formation from isoprene through its separate contributions to ALW, particle acidity, and aqueous chemistry (Nguyen et al., 2014; Surratt et al., 2010; Xu et al., 2015). Laboratory studies have not yet elucidated the role of each factor in the reversibility of isoprene SOA, and we do not have sufficient sulfate data to characterize such effects with our analysis. However, it is worth noting that particle acidity is not likely a factor in the relative split between reversible and irreversible aqSOA formed from isoprene. Studies predict that particles in the eastern U.S. are highly acidic throughout the year (Battaglia et al., 2017; Guo et al., 2016; Guo et al., 2015; Weber et al., 2016), and acidity is not a limiting factor in isoprene SOA formation during the summer (Budisulistiorini et al., 2016; Xu et al., 2015). The implication from our observations is that reversible aqSOA from isoprene forms even in the presence of such persistently acidic particles. This further questions the treatment of isoprene SOA as an irreversible uptake process in models."

**Minor Comments**

*7. Page 1, line 31. Oxford comma is recommended here after glyoxal.*

Comma has been added, as suggested.

*8. Page 2, line 1. Besides all the literature the author listed here, I believe Riedel et al., (ES&T Letters, 2015) should also be included as well when talking about reactive uptake of IEPOX.*

We have added the Riedel et al. (2015) reference, as suggested.

*9. Page 2, line 14. The author used an incorrect example here. Oligomerization is a non-reversible process, as also shown in De Haan et al. that the author cited.*

We respectfully disagree with the Referee on this point. We agree that De Haan et al. (2009) show oligomerization pathways that are irreversible (e.g., their Scheme 1). However, they also observe significant evaporation of glyoxal and methylglyoxal from aqueous droplets that undergo drying. This was likely due to hydrated forms of each compound, as well as dimers and trimers formed from self-reactions. See also the recent review article on aqSOA by McNeill (2015) which describes oligomerization of glyoxal and methylglyoxal as reversible.

**10. Page 2, line 15. Oxford comma is recommended here after inorganics.**

Comma has been added, as suggested.

**11. Page 3, line 25. Because the author performed the experiment using a home-built mist chamber, is there any characterization of this mist chamber, such as recovery efficiency of the gas phase species? Such information would help the reader in understanding the performance of the mist chamber and also error bar of the measurement.**

We have performed a detailed characterization of the MC, which is the topic of a manuscript in preparation. We cannot reference this work, in accord with ACP guidelines. However, this MC has been used in many prior studies (e.g., (Anderson et al., 2008; El-Sayed et al., 2016; 2015; Ervens et al., 2011; Hennigan et al., 2008; Hennigan et al., 2009; Zhang et al., 2012). In both of the El-Sayed et al. papers, we present experimental schematics and include detailed discussion of the MC, including LODs and measurement uncertainty. We have also added the following sentence to the Methods section: "The MC and $WSOC_g$ measurement have been detailed elsewhere (El-Sayed et al., 2016; 2015)."

**12. Page 5, line 10. Has the author compared the ambient RH of summer and winter times? Does ambient RH have an effect on $WSOC_{p,\,dry}/WSOC_p$**

Ambient RH does have an effect on $WSOC_{p,dry}/WSOC_p$ during summer (El-Sayed et al., 2016) and late spring, but not during other seasons (El-Sayed et al., 2015). The ambient RH was generally similar across the seasons; this is a point of discussion in our manuscript in preparation.

**13. Page 5, line 33-34. As previously mentioned, it would be better if the author could compare other BVOCs with WSOC obtained from this study to rule out the possibilities of other BVOCs producing WSOC.**

See our response to comment #2 above.

**14. Page 7, line 21. Has the author examined the relationship between ozone and**

*WSOC$_p$? If there is a correlation, then it means other BVOCs can also contribute to WSOC$_p$ as well.*

See our response to comment #2 above.

*15. Page 7, line 33-35. The sentence "If isoprene is indeed. . ." seems to be out of the place here because it does not go with the sentence below logically. The author can either elaborate more on this sentence or delete this sentence.*

We agree with the Referee's suggestion and have moved the order of these sentences so they now read: "If isoprene is indeed associated with the evaporated WSOC$_p$ that we observed during the late spring and summer, then a logical question is why we did not observe this phenomenon during measurements throughout September (Fig. 1, El-Sayed et al. (2015)). Although isoprene emissions decrease dramatically during September, there are still periods with elevated concentrations."

*16. Page 8, line 22. AOD was not defined previously. Please define.*

AOD has been defined.

*17. Page 8, line 23-line 36. This paragraph is pretty confusing because there are different concepts and ideas intertwined with each other. The author can talk about the results in Fig. 1 first, and then mention Liu et al. and Riva et al., and lastly talk about McNeill et al.*

We have revised this entire paragraph for clarity so that it now reads: "The effect of water evaporation on WSOC$_p$ also has important implications for the representation of SOA formation in models. The results in Fig. 1 show that ≈10 to 15 % of the total WSOC$_p$ evaporates with drying during the late spring and summer, on average. This suggests that the fraction of aqSOA that is formed reversibly is much higher than 15% since the measurement of WSOC$_p$ includes compounds formed through uptake of volatile water-soluble organic gases to aerosol water (aqSOA) and compounds formed through traditional SOA partitioning (e.g., (Donahue et al., 2009)). Further, the fraction of WSOC$_p$ that evaporates with drying is variable, with values of up to 60 % for individual measurements (El-Sayed et al., 2016). Together, these results indicate that representations of aqSOA formation through irreversible uptake schemes are not consistent with actual atmospheric phenomena. Models that include aqSOA and aerosol multiphase chemistry can improve predictions of OA (e.g., (Carlton et al., 2008; Marais et al., 2016)). A complication of model evaluations is that comparisons of modeled OA concentrations to ambient measurements may be problematic if the measurements, themselves, are subject to the bias discussed above. For this reason, accounting for both reversible and irreversible uptake of water-soluble organic gases to liquid water is critical (McNeill, 2015). Consistent with laboratory studies (Faust et al., 2017), our observations suggest that treatment of aqSOA as an irreversible uptake process is not consistent with actual phenomena occurring in the atmosphere, especially in the eastern U.S. Although likely due to a different mechanism, Liu et al. (2016) and Riva et

al. (2017) also showed that isoprene oxidation forms semi-volatile compounds that re-partition back to the gas phase after forming SOA."

**18. Page 8, line 32-line 33. What are the traditional SOA pathways? I would recommend specifying it more clearly because "uptake to aqueous particles" sounds like a traditional pathway of SOA formation to me as well.**

We have clarified the text so that it now reads: "This suggests that the fraction of aqSOA that is formed reversibly is much higher than 15%, since the measurement of $WSOC_p$ includes compounds formed through uptake of volatile water-soluble organic gases to aerosol water (aqSOA) and compounds formed through traditional SOA partitioning (e.g., (Donahue et al., 2009))."

**19. Page 8, line 36. Perhaps I am missing something here. Why is the reversible aqSOA even higher given that ~10 to 15% of the total $WSOC_p$ evaporates?**

We have clarified this sentence so that it now reads: "The results in Fig. 1 show that ≈10 to 15 % of the total $WSOC_p$ evaporates with drying during the late spring and summer, on average.  This suggests that the fraction of aqSOA that is formed reversibly is much higher than 15% since the measurement of $WSOC_p$ includes compounds formed through uptake to aqueous particles (aqSOA) and compounds formed through traditional SOA pathways."

**20. Page 9, line 15. The reason that ambient IEPOX-SOA has a low volatility can additionally be attributed to the higher viscosity of SOA. If the viscosity of the SOA is higher, then it will be more difficult for the semi-volatile species to evaporate and escape from the particle phase within the timescale of the measurement, as discussed in Vaden et al. (2011) and Zhang et al. (2015). This can be another reason why some ambient aerosols do not show strong reversibility and the author should consider putting it in the discussion.**

The reviewer brings up an interesting point.  However, we do not believe that higher viscosity affects the apparent volatility of IEPOX-SOA in the studies we have referenced.  During the summertime in the eastern U.S., relative humidity is high and aerosol liquid water is typically abundant, especially at night (e.g., (Guo et al., 2015)).  This leads to conditions where the SOA has a liquid phase state (Shiraiwa et al., 2017), and is not phase-separated from ALW (Pye et al., 2017).  Extensive results from the groups of Scot Martin and Allan Bertram show that under such conditions, viscosity of the SOA should not lead to diffusion limitations and longer equilibration timescales.

**21. Page 10, line 13-line 21. This paragraph is confusing as well. At the beginning of the paragraph the author seems to believe the effect of ALW on WSOC is not as significant as OA concentration. By the end of the paragraph the author concludes that WSOC is not due to OA partitioning. Please revise this paragraph to give a clearer explanation.**

We have clarified this paragraph so that it now reads: "The physical properties that affect SOA formed through absorptive partitioning (what Ervens et al. (2011) call gasSOA) and SOA formed through an aqueous mediated pathway (aqSOA) are fundamentally different (vapor pressure and gas solubility in water, respectively). Note that ALW can affect SOA formed through traditional absorptive partitioning by increasing the total concentration and decreasing the average molecular weight of the absorbing OM phase (Seinfeld and Pankow, 2003). Models predict that this phenomenon enhances SOA concentrations in the eastern U.S. (Jathar et al., 2016; Pankow et al., 2015), and that drying the particles will result in the evaporation of some semi-volatile SOA compounds in response to this perturbation (Pankow, 2010). However, we believe that the observed $WSOC_p$ evaporation during the late spring and summer seasons was the result of reversible aqSOA. The effect of ALW on gas-particle partitioning is more pronounced at low organic concentrations (1 to 2 $\mu g\ m^{-3}$), and its sensitivity becomes less profound at higher OA concentrations (Pankow, 2010). Previous results from our group showed that the evaporated $WSOC_p$ concentrations increased significantly with an increase in OA concentrations (El-Sayed et al., 2016). Further, the semi-volatile organic compounds most influenced by this water effect are predicted to be the less oxidized, fresh SOA (Pankow, 2010). $WSOC_p$ is much more strongly correlated with the LV-OOA (low-volatility oxygenated organic aerosol) factor identified by the AMS compared to the SV-OOA (semi-volatile OOA) factor (Kondo et al., 2007; Sun et al., 2011; Xu et al., 2017). This suggests that the evaporation of $WSOC_p$ was not due to the overall effects on OA partitioning (Jathar et al., 2016), but was due to the reversible partitioning of water-soluble organic gases to aerosol water. In the following sections, we characterize the reasons underlying the seasonal differences in $WSOC_{p,dry}/WSOC_p$ shown in Fig. 1."

*22. Page 10, line 18. Please define LV-OOA and SV-OOA before using these two terms.*

These have been defined.

**Cited References**

Anderson, C.H., Dibb, J.E., Griffin, R.J., Hagler, G.S.W., Bergin, M.H., Atmospheric water-soluble organic carbon measurements at Summit, Greenland, *Atmospheric Environment* **42**(2008), pp. 5612-5621.

Anderson, D.C. *et al.*, Measured and modeled CO and NO y in DISCOVER-AQ: An evaluation of emissions and chemistry over the eastern US, *Atmospheric environment* **96**(2014), pp. 78-87.

Battaglia Jr, M.A., Douglas, S., Hennigan, C.J., Effect of the Urban Heat Island on Aerosol pH, *Environmental science & technology* **51**(2017), p. 13095.

Beyersdorf, A.J. *et al.*, The impacts of aerosol loading, composition, and water uptake on aerosol extinction variability in the Baltimore–Washington, D.C. region, *Atmos. Chem. Phys.* **16**(2016), pp. 1003-1015.

Budisulistiorini, S.H. *et al.*, Seasonal characterization of submicron aerosol chemical composition and organic aerosol sources in the southeastern United States: Atlanta, Georgia,and Look Rock, Tennessee, *Atmos. Chem. Phys.* **16**(2016), pp. 5171-5189.

Carlton, A.G. *et al.*, CMAQ Model Performance Enhanced When In-Cloud Secondary Organic Aerosol is Included: Comparisons of Organic Carbon Predictions with Measurements, *Environmental Science & Technology* **42**(2008), pp. 8798-8802.

Chan Miller, C. *et al.*, Glyoxal yield from isoprene oxidation and relation to formaldehyde: chemical mechanism, constraints from SENEX aircraft observations, and interpretation of OMI satellite data, *Atmos. Chem. Phys.* **17**(2017), pp. 8725-8738.

De Haan, D.O. *et al.*, Secondary Organic Aerosol Formation by Self-Reactions of Methylglyoxal and Glyoxal in Evaporating Droplets, *Environmental Science & Technology* **43**(2009), pp. 8184-8190.

Ding, X. *et al.*, Spatial and seasonal trends in biogenic secondary organic aerosol tracers and water-soluble organic carbon in the southeastern United States, *Environmental science & technology* **42**(2008), pp. 5171-5176.

Donahue, N.M., Robinson, A.L., Pandis, S.N., Atmospheric organic particulate matter: From smoke to secondary organic aerosol, *Atmospheric Environment* **43**(2009), pp. 94-106.

El-Sayed, M.M.H., Amenumey, D., Hennigan, C.J., Drying-Induced Evaporation of Secondary Organic Aerosol during Summer, *Environmental Science & Technology* **50**(2016), pp. 3626-3633.

El-Sayed, M.M.H., Wang, Y.Q., Hennigan, C.J., Direct atmospheric evidence for the irreversible formation of aqueous secondary organic aerosol, *Geophysical Research Letters* **42**(2015), pp. 5577-5586.

Ervens, B., Turpin, B.J., Weber, R.J., Secondary organic aerosol formation in cloud droplets and aqueous particles (aqSOA): a review of laboratory, field and model studies, *Atmos. Chem. Phys.* **11**(2011), pp. 11069-11102.

Faust, J.A., Wong, J.P.S., Lee, A.K.Y., Abbatt, J.P.D., Role of Aerosol Liquid Water in Secondary Organic Aerosol Formation from Volatile Organic Compounds, *Environmental Science & Technology* **51**(2017), pp. 1405-1413.

Guenther, A.B. *et al.*, The Model of Emissions of Gases and Aerosols from Nature version 2.1 (MEGAN2.1): an extended and updated framework for modeling biogenic emissions, *Geosci. Model Dev.* **5**(2012), pp. 1471-1492.

Guo, H. *et al.*, Fine particle pH and gas–particle phase partitioning of inorganic species in Pasadena, California, during the 2010 CalNex campaign, *Atmospheric Chemistry and Physics* **17**(2017), pp. 5703-5719.

Guo, H. *et al.*, Fine particle pH and the partitioning of nitric acid during winter in the northeastern United States, *Journal of Geophysical Research: Atmospheres* **121**(2016).

Guo, H. *et al.*, Fine-particle water and pH in the southeastern United States, *Atmos. Chem. Phys.* **15**(2015), pp. 5211-5228.

Hennigan, C.J., Bergin, M.H., Dibb, J.E., Weber, R.J., Enhanced secondary organic aerosol formation due to water uptake by fine particles, *Geophysical Research Letters* **35**(2008).

Hennigan, C.J., Bergin, M.H., Russell, A.G., Nenes, A., Weber, R.J., Gas/particle partitioning of water-soluble organic aerosol in Atlanta, *Atmospheric Chemistry and Physics* **9**(2009), pp. 3613-3628.

Jathar, S.H., Cappa, C.D., Wexler, A.S., Seinfeld, J.H., Kleeman, M.J., Simulating secondary organic aerosol in a regional air quality model using the statistical oxidation model – Part 1: Assessing the influence of constrained multi-generational ageing, *Atmos. Chem. Phys.* **16**(2016), pp. 2309-2322.

Jimenez, J.L. *et al.*, Evolution of Organic Aerosols in the Atmosphere, *Science* **326**(2009), p. 1525.

Khlystov, A., Stanier, C.O., Takahama, S., Pandis, S.N., Water content of ambient aerosol during the Pittsburgh air quality study, *Journal of Geophysical Research-Atmospheres* **110**(2005).

Kleindienst, T.E. *et al.*, Estimates of the contributions of biogenic and anthropogenic hydrocarbons to secondary organic aerosol at a southeastern US location, *Atmospheric Environment* **41**(2007), pp. 8288-8300.

Kondo, Y. *et al.*, Oxygenated and water-soluble organic aerosols in Tokyo, *Journal of Geophysical Research-Atmospheres* **112**(2007).

Liu, Y., Kuwata, M., McKinney, K.A., Martin, S.T., Uptake and release of gaseous species accompanying the reactions of isoprene photo-oxidation products with sulfate particles, *Physical Chemistry Chemical Physics* **18**(2016), pp. 1595-1600.

Marais, E.A. *et al.*, Aqueous-phase mechanism for secondary organic aerosol formation from isoprene: application to the southeast United States and co-benefit of SO2 emission controls, *Atmos. Chem. Phys.* **16**(2016), pp. 1603-1618.

McNeill, V.F., Aqueous Organic Chemistry in the Atmosphere: Sources and Chemical Processing of Organic Aerosols, *Environmental Science & Technology* **49**(2015), pp. 1237-1244.

Nguyen, T.B. *et al.*, Organic aerosol formation from the reactive uptake of isoprene epoxydiols (IEPOX) onto non-acidified inorganic seeds, *Atmos. Chem. Phys.* **14**(2014), pp. 3497-3510.

Pankow, J.F., Organic particulate material levels in the atmosphere: Conditions favoring sensitivity to varying relative humidity and temperature, *Proceedings of the National Academy of Sciences* **107**(2010), pp. 6682-6686.

Pankow, J.F. *et al.*, Molecular view modeling of atmospheric organic particulate matter: Incorporating molecular structure and co-condensation of water, *Atmospheric Environment* **122**(2015), pp. 400-408.

Pye, H.O.T. *et al.*, On the implications of aerosol liquid water and phase separation for organic aerosol mass, *Atmos. Chem. Phys.* **17**(2017), pp. 343-369.

Pye, H.O.T. *et al.*, Epoxide Pathways Improve Model Predictions of Isoprene Markers and Reveal Key Role of Acidity in Aerosol Formation, *Environmental Science & Technology* **47**(2013), pp. 11056-11064.

Riedel, T.P. *et al.*, Heterogeneous Reactions of Isoprene-Derived Epoxides: Reaction Probabilities and Molar Secondary Organic Aerosol Yield Estimates, *Environmental Science & Technology Letters* **2**(2015), pp. 38-42.

Riva, M. *et al.*, Multiphase reactivity of gaseous hydroperoxide oligomers produced from isoprene ozonolysis in the presence of acidified aerosols, *Atmospheric Environment* **152**(2017), pp. 314-322.

Seinfeld, J.H., Pankow, J.F., Organic atmospheric particulate material, *Annual review of physical chemistry* **54**(2003), pp. 121-140.

Shiraiwa, M. *et al.*, Global distribution of particle phase state in atmospheric secondary organic aerosols, *Nature communications* **8**(2017).

Sun, Y.L. *et al.*, Characterization of the sources and processes of organic and inorganic aerosols in New York city with a high-resolution time-of-flight aerosol mass apectrometer, *Atmos. Chem. Phys.* **11**(2011), pp. 1581-1602.

Surratt, J.D. *et al.*, Reactive intermediates revealed in secondary organic aerosol formation from isoprene, *Proceedings of the National Academy of Sciences* **107**(2010), pp. 6640-6645.

Timonen, H. *et al.*, Characteristics, sources and water-solubility of ambient submicron organic aerosol in springtime in Helsinki, Finland, *Journal of Aerosol Science* **56**(2013), pp. 61-77.

Weber, R.J., Guo, H., Russell, A.G., Nenes, A., High aerosol acidity despite declining atmospheric sulfate concentrations over the past 15 years, *Nature Geoscience* **9**(2016), pp. 282-285.

Weber, R.J. *et al.*, A study of secondary organic aerosol formation in the anthropogenic-influenced southeastern United States, *Journal of Geophysical Research-Atmospheres* **112**(2007).

Xu, L. *et al.*, Effects of anthropogenic emissions on aerosol formation from isoprene and monoterpenes in the southeastern United States, *Proceedings of the National Academy of Sciences* **112**(2015), pp. 37-42.

Xu, L., Guo, H., Weber, R.J., Ng, N.L., Chemical characterization of water-soluble organic aerosol in contrasting rural and urban environments in the southeastern United States, *Environmental science & technology* **51**(2016), pp. 78-88.

Xu, W. *et al.*, Effects of Aqueous-Phase and Photochemical Processing on Secondary Organic Aerosol Formation and Evolution in Beijing, China, *Environmental Science & Technology* **51**(2017), pp. 762-770.

Yu, K. *et al.*, Sensitivity to grid resolution in the ability of a chemical transport model to simulate observed oxidant chemistry under high-isoprene conditions, *Atmos. Chem. Phys.* **16**(2016), pp. 4369-4378.

Zhang, X.L. *et al.*, On the gas-particle partitioning of soluble organic aerosol in two urban atmospheres with contrasting emissions: 1. Bulk water-soluble organic carbon, *Journal of Geophysical Research-Atmospheres* **117**(2012).

[revised manuscript text omitted]
 is a scatter plot of NOx concentrations measured at two different sites in the Baltimore region. Figure S2 presents seasonal daytime and nighttime $WSOC_{p,dry}/WSOC_p$ ratios. Figure S3 depicts the summertime diurnal profiles of isoprene, and $WSOC_g$ concentrations shifted by 3 h. Figure S4 shows boxplots of $WSOC_p$ and evaporated $WSOC_p$ concentrations as a function of isoprene concentrations at 9 h time delay relative to isoprene concentrations. Figure S5 illustrates the median evaporated $WSOC_p$ concentrations as a function of $WSOC_g$ concentrations at different time delays relative to $WSOC_g$ concentrations, and a scatter and box plot corresponding to one time lag (4 h). Finally, Figure S6 shows scatter and boxplots of $WSOC_g$ and evaporated WSOCp concentrations as a function of isoprene concentrations at 3 h and 9-h time lags, respectively, during the summer.

**Table S1.** Comparison of ambient RH and RH sampled through the silica gel dryer across the different seasons.

| Ambient RH (%) | RH-through dryer Mean ± 1σ (%) | | | |
|---|---|---|---|---|
| | Fall | Winter | Spring | Summer |
| 20 | | | 10.7 ± 0.5 | |
| 30 | | | 19.8 ± 0.4 | |
| 40 | | | 21.1 ± 0.5 | 32.7 ± 0.7 |
| 50 | | 15.5 ± 0.2 | 16.5 ± 0.3 | 35.0 ± 0.9 |
| 60 | 46.0 ± 0.7 | 19.2 ± 0.8 | 31.4 ± 0.3 | 36.6 ± 0.4 |
| 70 | 42.3 ± 1.9 | 20.8 ± 0.7 | 32.0 ± 0.4 | 35.1 ± 0.1 |
| 80 | 42.5 ± 0.5 | 22.8 ± 0.9 | 22.5 ± 0.6 | 40.0 ± 0.1 |
| 90 | 42.2 ± 1.2 | 23.6 ± 0.8 | | |

[Figure]

**Figure S1:** Scatter plot of daily average $NO_x$ concentrations (in ppb or nmol mol$^{-1}$) at the HU-Beltsville and Essex sites for one year. The solid black line is the linear fit based on a least-squares regression analysis.

[Figure]

**Figure S2:** Daytime (brown) and nighttime (blue) seasonal $WSOC_{p,dry}/WSOC_p$ ratios. Circles and diamonds represent the daytime and nighttime averages, respectively. The green dotted line at unity is for visual reference.

[Figure]

**Figure S3:** Summertime average diurnal profiles of isoprene concentrations (in ppbC or nmol mol$^{-1}$, green circles), and WSOC$_g$ concentrations shifted 3 h prior to their measurement (blue diamonds). All concentrations pertain to the summer, ozone season (starting from early June until late August of 2015) when hourly isoprene measurements were available. Error bars represent ±1σ. The two series were offset by 0.3 h for clarity.

[Figure]

**Figure S4:** Boxplots of WSOC$_p$ and evaporated WSOC$_p$ concentrations as a function of isoprene concentrations (in ppbC or nmol mol[-1]) at 9 h time delay (n = 9 h) relative to isoprene concentrations. Blue circles and red diamonds represent the means of the WSOC$_p$ and evaporated WSOC$_p$ concentrations at each isoprene bin, respectively. Note that the 95[th] percentile of the evaporated WSOC$_p$ concentration for the highest isoprene bin (> 5 ppbC, or 5 nmol mol[-1]) is off scale (7.6 μg m[-3]).

[Figure]

**Figure S5:** (a) Median evaporated WSOC$_p$ concentrations as a function of WSOC$_g$ concentrations at different time delays relative to the WSOC$_g$ concentrations. (b) Scatter and box plot (median, inter quartile range, and 5[th] and 95[th] percentiles) of the evaporated WSOC$_p$ concentration at 4-h time lag vs. WSOC$_g$. Red circles represent the mean of each bin. Note that approximately 2% of the individual measurements are off scale.

[Figure]

**Figure S6:** Scatter and boxplot of $WSOC_g$ concentrations with a 3-h time lag (top) and the evaporated $WSOC_p$ concentrations with 9-h time lag (bottom) as a function of isoprene concentrations (in ppbC, or nmol mol[-1]) during the summer. The following isoprene concentrations bins were defined: $< 1, 1 − 2, 2 − 3, 3 − 4, 4 − 5$, and $> 5$ ppbC (nmol mol[-1]). For each bin, mean (red marker), median (horizontal black line), 25th and 75th percentiles (lower and upper box values), as well as 5th and 95th percentiles (vertical lines) are shown. Note ≈1% of the data are off scale in each figure.